# CONDITIONAL-$t^3$VAE: EQUITABLE LATENT SPACE ALLOCATION FOR FAIR GENERATION

## ABSTRACT

Variational Autoencoders (VAEs) with global priors mirror the training set's class frequency in latent space, underrepresenting tail classes and reducing generative fairness on imbalanced datasets. While $t^3$VAE improves robustness via heavy-tailed Student's t-distribution priors, it still allocates latent volume proportionally to the class frequency. In this work, we address this issue by explicitly enforcing equitable latent space allocation across classes. To this end, we propose Conditional-$t^3$VAE, which defines a per-class Student's t joint prior over latent and output variables, preventing dominance by majority classes. Our model is optimized using a closed-form objective derived from the $\gamma$-power divergence. Moreover, for class-balanced generation, we derive an equal-weight latent mixture of Student's t-distributions. On SVHN-LT, CIFAR100-LT, and CelebA, Conditional-$t^3$VAE consistently achieves lower FID scores than both $t^3$VAE and Gaussian-based VAE baselines, particularly under severe class imbalance. In per-class F1 evaluations, Conditional-$t^3$VAE also outperforms the conditional Gaussian VAE across all highly imbalanced settings. While Gaussian-based models remain competitive under mild imbalance ratio ($\rho < 5$), our approach substantially improves generative fairness and diversity in more extreme regimes.

## 1 INTRODUCTION

Class imbalance and long-tail distributions are prevalent in real-world datasets, yet generative models often fail to represent rare classes accurately. When trained on skewed data, these models tend to overfit dominant modes and underrepresent minority ones in latent and output spaces, resulting in biased or unfair generations. This issue is especially critical in sensitive applications such as facial synthesis (Mehta et al., 2024) and medical imaging (Pinaya et al., 2022), where such biases can exacerbate social and diagnostic disparities (Naik & Nushi, 2023).

Variational Autoencoders (VAEs) (Kingma & Welling, 2013) are a widely used class of generative models, valued for their probabilistic formulation, stable training, and compatibility with latent-variable modeling frameworks leading in image quality such as Latent Diffusion Models (LDMs) (Rombach et al., 2022). Although GANs and diffusion models often achieve lower FID scores, VAEs offer unique advantages in class-conditional generation, interpretability, and efficient inference, making them strong candidates for improving fairness under class imbalance. Standard VAEs commonly use isotropic Gaussian priors, which inadequately model heavy-tailed structures and rare phenomena (Tam & Dunson, 2025). Prior efforts to address this have introduced non-Gaussian priors, particularly Student's t-distributions (Takahashi et al., 2018; Abiri & Ohlsson, 2020; Eguchi, 2021; Kim et al., 2024), to enhance robustness. However, these approaches often rely on global priors, causing the latent space to be dominated by majority classes under skewed distributions.

We address this issue with Conditional-$t^3$VAE (C-$t^3$VAE), a conditional generative model that imposes a per-class Student's t-distribution prior over the joint latent-output space. This design ensures allocating an equal latent space volume per class, thereby mitigating majority class dominance, while the heavy tails of the Student's t-distribution more effectively capture intra-class variation. To enable class-balanced sampling, we introduce an equal-weight mixture of Student's t-distributions with analytically derived component variances. Together, these components enable balanced class-conditional generation and mitigate bias present in unconditional models. We summarize our main theoretical and empirical contributions in the following points :

- We propose the C-$t^3$VAE model with a training objective based on the $\gamma$-power divergence.
- We develop an equal-weight latent mixture sampling scheme with analytically derived optimal variance scaling for each component.
- We outperform relevant baselines in FID on SVHN-LT (Netzer et al., 2011), CIFAR100-LT (Cao et al., 2019), and CelebA (Liu et al., 2015) under severe imbalance, and show via per-class evaluation that C-$t^3$VAE better avoids mode collapse, exceeding a conditional VAE in per class Recall and F1 while remaining competitive on Precision.
- We identify the imbalance ratio threshold $\rho \approx 5$, beyond which Gaussian priors become suboptimal, providing guidance for model selection on skewed datasets.

## 2 RELATED WORK

Since the introduction of VAEs (Kingma & Welling, 2013), many extensions have sought to improve latent representation by replacing the standard Gaussian prior with more expressive alternatives. These include Gaussian mixtures (Saseendran et al., 2021; Dilokthanakul et al., 2016), hyperspherical priors (Davidson et al., 2018), normalizing flows (Jaini et al., 2020), Riemannian priors (Chadebec et al., 2023), and implicit distributions (Takahashi et al., 2019). While most retain the ELBO formulation, others adopt alternative objectives or divergence measures for added flexibility.

To address long-tailed or imbalanced data, Student's t-distributions have been explored for their robustness and heavy tails (Tam & Dunson, 2025). Methods such as (Takahashi et al., 2018) and (Abiri & Ohlsson, 2020) model the latent space of the autoencoder through a t-distributed prior, and rely on KL-divergence-based ELBO objectives. However, since the KL divergence lacks a closed-form solution for the Student's t-distributions, these methods resort to numerical approximation. The $t^3$VAE (Kim et al., 2024) improves on this by modeling the joint latent-output distribution and optimizing a closed-form $\gamma$-divergence objective (Eguchi, 2021). Nonetheless, it still employs a global latent prior, which results in a latent space volume allocation reflecting class frequency and leading to imbalance present in the generated samples.

Other works address fairness in generative modeling using normalizing flows (Jaini et al., 2020) and diffusion models (Qin et al., 2023; Pandey et al., 2025). However, since leading image generation models are based on Latent Diffusion Models (Rombach et al., 2022) and VAEs are a corner-stone in the design of these models, we deem it crucial to improve VAEs' ability to handle imbalanced data. Consequently, as fairness in latent space allocation remains under-explored in VAEs, we introduce a class-conditional, heavy-tailed prior to address this gap and allow for balanced latent space allocation across classes.

## 3 BACKGROUND

This section introduces the theoretical background and baseline models relevant to our work. We assume access to a labeled, imbalanced dataset $\mathcal{D} = \{(x_i, y_i)\}_{i=1}^N$, where $x_i \in \mathbb{R}^n$ is a data sample of dimension $n$, $y_i \in \{1, \ldots, K\}$ its class label and $m$ the latent space dimension.

### 3.1 VAEs AND CONDITIONAL-VAEs

VAEs (Kingma & Welling, 2013) are generative models trained via variational inference by maximizing the Evidence Lower Bound (ELBO) of the log-likelihood. The standard objective of this model is

$$\mathcal{L}_{\theta,\phi} := \mathbb{E}_{z \sim q_\phi(\cdot|x)}[\log p_\theta(x|z)] - \mathcal{D}_{KL}(q_\phi(z|x)\|p(z)), \tag{1}$$

where the first term is the reconstruction loss with $p_\theta(x|z)$ being the decoder model. The second term is the Kullback–Leibler (KL) divergence between the approximate posterior $q_\phi(z|x)$ and the prior $p(z)$. The $\beta$-VAE is a weighted variant of the VAE model which introduces a $\beta$ scaling term for the KL divergence (Higgins et al., 2017):

$$\mathcal{L}_{\theta,\phi} := \mathbb{E}_{z \sim q_\phi(\cdot|x)}[\log p_\theta(x|z)] - \beta \mathcal{D}_{KL}(q_\phi(z|x)\|p(z)), \tag{2}$$

with $p(z) \sim \mathcal{N}_m(0, I)$, $q_\phi(\cdot|x) \sim \mathcal{N}_m(\mu_\phi(x), \Sigma_\phi(x))$, and $p_\theta(x|z) \sim \mathcal{N}_m(\mu_\theta(z), \sigma^2 I)$. $\mu_\phi(\cdot)$ and $\Sigma_\phi(\cdot)$ are the mean and the covariance matrices inferred through a neural network with parameter $\phi$

given the input $x$. Moreover, $\mu_\theta(\cdot)$ is the decoder neural network with parameter $\theta$ and $\sigma$ is a parameter controlling the decoder's output covariance. This variant of the VAE model allows to place more weight on disentangling the latent space or on the reconstruction of the data points. To generate samples from the VAE or the $\beta$-VAE model, we sample a latent vector $z \sim \mathcal{N}_m(0, I)$. Then, the generated data point would be $\hat{x} \sim \mathcal{N}_m(\mu_\theta(z), \sigma^2 I)$.

Nevertheless, since Eq. (1) and Eq. (2) optimize the ELBO over the data distribution $p_{data}(x)$, which can be decomposed as $p_{\text{data}}(x) = \sum_{y_i} p(y_i) \, p_{\text{data}}(x \mid y_i)$, this optimization inherently biases the model toward head classes with larger $p(y_i)$. As a result, most generated samples come from overrepresented classes, while tail classes' samples are underrepresented and of lower quality, a phenomenon commonly referred to as *mode collapse*. Therefore, when labels are available, it is preferable to define class-conditional posterior and prior distributions: $q_\phi(z|x, y)$ and $p(z|y)$. This yields the Conditional-VAE (CVAE) model trained using the objective (Kingma et al., 2014):

$$\sum_y \mathbb{E}_{z \sim q_\phi(\cdot|x,y)}[\log p_\theta(x|z, y)] - \beta \mathcal{D}_{KL}(q_\phi(z|x, y)\|p(z|y)). \tag{3}$$

Here, in Eq. (3) we constrain all $p(y_i)$ to be equal and omit them from the loss function in order not to exacerbate the issue of class imbalance in the latent space. Also, we define $p(z|y) \sim \mathcal{N}_m(\mu_y, I)$ with learnable class-wise means $\mu_y$. To generate a data point $\hat{x}_y$ from class $y$, we sample $z_y \sim \mathcal{N}_m(\mu_y, I)$, then we get $\hat{x}_y \sim p_\theta(x|z_y, y)$. Nevertheless, despite conditioning, this formulation remains Gaussian. Unlike Student's t-distributions, Gaussian priors poorly approximate heavy-tailed data distributions (Tam & Dunson, 2025).

### 3.2 Multivariate Student's t-Distribution and $\gamma$-Power Divergence

A $d$-dimensional Student's t-distribution with mean $\mu \in \mathbb{R}^d$, covariance $\Sigma \in \mathbb{R}^{d \times d}$, and degrees of freedom $\nu > 2$ is a heavy-tail, super-Gaussian distribution defined as

$$t_d(x) = C_{\nu,d}|\Sigma|^{-\frac{1}{2}} \left(1 + \frac{(x-\mu)^\top \Sigma^{-1}(x-\mu)}{\nu}\right)^{-\frac{\nu+d}{2}}, \qquad C_{\nu,d} = \frac{\Gamma\left(\frac{\nu+d}{2}\right)}{\Gamma\left(\frac{\nu}{2}\right)(\nu\pi)^{\frac{d}{2}}}. \tag{4}$$

The power form of this distribution prevents a closed-form KL divergence between two Student's t-distributions. Instead, the $\gamma$-power divergence $\mathcal{D}_\gamma(q\|p)$ is used (Eguchi, 2021; Kim et al., 2024). This divergence is defined for $q \sim t_d(\mu_0; \Sigma_0; \nu)$, $p \sim t_d(\mu_1; \Sigma_1; \nu)$ starting from the $\gamma$-entropy $\mathcal{H}_\gamma(p)$ and cross-entropy $\mathcal{C}_\gamma(q, p)$

$$\mathcal{H}_\gamma(p) := -\|p\|_{1+\gamma} = -\left(\int p(x)^{1+\gamma} dx\right)^{\frac{1}{1+\gamma}}, \qquad \mathcal{C}_\gamma(q, p) := -\int q(x) \left(\frac{p(x)}{\|p\|_{1+\gamma}}\right)^\gamma dx.$$

$$\mathcal{D}_\gamma(q\|p) := \gamma^{-1}\left(\mathcal{C}_\gamma(q, p) - \mathcal{H}_\gamma(p)\right) \tag{5}$$

with $\gamma = -\frac{2}{\nu+d}$. Then, substituting the definition of a Student's t-distribution from Eq. (4) into Eq. (5), the following closed-form formula for the $\gamma$-power divergence can be derived (See Appendix A):

$$\mathcal{D}_\gamma(q\|p) = -\frac{C_{\nu,d}^{\frac{\gamma}{1+\gamma}}}{\gamma}\left(1 + \frac{d}{\nu-2}\right)^{-\frac{\gamma}{1+\gamma}}\left[-|\Sigma_0|^{-\frac{\gamma}{2(1+\gamma)}}\left(1 + \frac{d}{\nu-2}\right)\right.$$

$$\left. + |\Sigma_1|^{-\frac{\gamma}{2}}|\Sigma_0|^{\frac{\gamma^2}{2(1+\gamma)}}\left(1 + \frac{\text{Tr}\left(\Sigma_1^{-1}\Sigma_0\right)}{\nu-2} + \frac{(\mu_0-\mu_1)^\top \Sigma_1^{-1}(\mu_0-\mu_1)}{\nu}\right)\right]. \tag{6}$$

### 3.3 $t^3$-Variational Autoencoder

#### 3.3.1 Definition

The $t^3$VAE model (Kim et al., 2024) is a non-ELBO-based autoencoder which models the joint prior distribution $p_\theta(x, z)$ using multivariate Student's t-distributions

$$p_\theta(x, z) \propto \sigma^{-n}\left[1 + \frac{1}{\nu}\left(\|z\|^2 + \frac{\|x - \mu_\theta(z)\|^2}{\sigma^2}\right)\right]^{-\frac{\nu+m+n}{2}}.$$

From this joint distribution, the marginal latent prior $p(z)$ and decoder distribution $p_\theta(x|z)$ can be defined. Furthermore, the posterior distribution is defined as :

$$q_\phi(z|x) = t_m\left(x \middle| \mu_\phi(x), \frac{\Sigma_\phi(x)}{1+\nu^{-1}n}, \nu+n\right).$$

Hence, the data-latent joint distribution would be $q_\phi(x,z) = p_{\text{data}}(x)q_\phi(z|x)$. As a result, relying on the $\gamma$-divergence in Eq. (6) applied to the $p_\theta(x,z)$ and $q_\phi(x,z)$ distributions, the following loss function is derived to optimize the $t^3$VAE's parameters :

$$\mathcal{L}_\gamma = \mathbb{E}_x\left[\frac{\mathbb{E}_z\left[\|x - \mu_\theta(z)\|^2\right]}{\sigma^2} + \|\mu_\phi(x)\|^2 + \frac{\nu\operatorname{Tr}(\Sigma_\phi(x))}{\nu+n-2} - \frac{\nu C_1}{C_2}|\Sigma_\phi(x)|^{-\frac{\gamma}{2(1+\gamma)}}\right], \quad (7)$$

with $\gamma = -\frac{2}{\nu+n+m}$ and $C_1$ and $C_2$ being theoretically derived constants. We note that the first term in this loss function represents the standard reconstruction term in VAE models and the rest of the terms are regularization terms over the latent space. To sample from the latent space of the $t^3$VAE, Kim et al. (2024) propose the $p_\nu^\star(z) = t_m(0, \tau^2 I, \nu+n)$ distribution with

$$\tau^2 = \frac{1}{1+\nu^{-1}n}\left(\frac{C_{\nu,n}}{\sigma^n} \cdot \frac{\nu-2}{\nu+n-2}\right)^{\frac{2}{\nu+n-2}}. \quad (8)$$

Moreover, sampling from a multi-variate Student's t-distribution $T \sim t_d(\mu, \Sigma, \nu)$ both in the learning (Eq. (7)) and sampling (Eq. (8)) phases is performed through the standard reparameteration trick for Student's t-distributions $T := \mu + Z\sqrt{\nu V^{-1}}$ where $Z \sim \mathcal{N}(0, \Sigma)$ and $V \sim \mathcal{X}^2(\nu)$.

### 3.3.2 $\beta$-$t^3$VAE

From Eq. (7) we can also define a $\beta$-$t^3$VAE model by multiplying all the regularization terms by a $\beta$ factor. Similarly to $\beta$-VAE models, this improves the versatility of the model and allows either a focus on generation or disentangling.

### 3.3.3 $\tau^2$ IMPROVEMENT

Closely analyzing the proposed derivation of $\tau^2$ of the $t^3$VAE model, we discovered a subtle issue in its mathematical formulation. The employed $\gamma$-power divergence presents a discrepancy to the correct formula in Eq. (6). We revised the formulation and the corrected $\tau^2$ is (See Appendix D):

$$\tau^2 = \frac{1}{1+\nu^{-1}n}\left(|\Sigma_\phi(x)|^{\frac{\gamma}{2}}\frac{C_{\nu,n}}{\sigma^n}\frac{\nu-2}{\nu+n-2}\right)^{\frac{2\gamma}{(1+\gamma)(2+\gamma m)}} \approx \frac{1}{1+\nu^{-1}n}\left(\frac{C_{\nu,n}}{\sigma^n}\frac{\nu-2}{\nu+n-2}\right)^{\frac{2\gamma}{2+\gamma m}}. \quad (9)$$

The corrected exact form of $\tau^2$ is applicable when $|\Sigma_\phi(x)|$ is known and when the dimension of the data is low. However, for high dimensional data, as handled in this work, we get $\gamma \approx 0$. Hence, one can use the approximation without any loss in accuracy. We note that the new form of $\tau^2$ leads to a similar empirical value of standard deviation compared to the previous form. Nevertheless, for correctness, in our sampling from the latent space of the $t^3$VAE model we use the approximation in Eq. (9).

In summary, although the $t^3$VAE effectively models heavy-tailed distributions through Student's t-distributions and $\gamma$-power divergence, it does not explicitly address class imbalance in the latent space by not allocating equal volume for each class. In the next section, we introduce a class-conditional variant of the $t^3$VAE, designed to ensure fair and balanced generation across all classes.

## 4 CONDITIONAL $t^3$-VARIATIONAL AUTOENCODER

We propose the Conditional $t^3$-Variational Autoencoder (C-$t^3$VAE), present its formulation, training objective, and sampling strategy. C-$t^3$VAE models the latent space as a mixture of Student's t-distributions, one per class, ensuring equal latent volume allocation and promoting fairness in generation. Intra-class imbalance is further handled by the heavy-tailed nature of the Student's t-distribution prior.

## 4.1 MODEL DEFINITION

The C-$t^3$VAE we propose is based on the following class conditional joint prior distribution

$$p_\theta(x, z|y) = \frac{C_{\nu,m+n}}{|\Sigma_x|^{\frac{1}{2}} |\Sigma_y|^{\frac{1}{2}}} \left[ 1 + \frac{(z - \mu_y)^\top \Sigma_y^{-1}(z - \mu_y) + (x - \mu_\theta(z))^\top \Sigma_x^{-1}(x - \mu_\theta(z))}{\nu} \right]^{-\frac{\nu+m+n}{2}}$$

with $\nu$, $n$ and $m$ being the degrees of freedom of the Student's t-distribution, the dimension of the input data and the dimension of the latent space respectively. $\mu_y \in \mathbb{R}^m$ is a learnable mean vector representing class centers in latent space of dimension $m$. Moreover, $\Sigma_x$ and $\Sigma_y$ are the covariance matrices of the prior distributions over the latent and output variables.

From this joint distribution, we can derive the conditional latent prior $p(z|y) = t_m(z|\mu_y, \Sigma_y, \nu)$ and decoder distribution (See Appendix B)

$$p_\theta(x|z, y) = t_n\left( x \middle| \mu_\theta(z), \frac{\left(1 + \nu^{-1}(z - \mu_y)^\top \Sigma_y^{-1}(z - \mu_y)\right)}{(1 + \nu^{-1}m)} \Sigma_x, \nu + m \right).$$

Furthermore, as in $t^3$VAE, we define the posterior $q_\phi(z|x)$ as a multivariate Student's t-distribution capturing heavy-tailed structure in the latent space :

$$q_\phi(z|x) = t_m\left( z \middle| \mu_\phi(x), \frac{\Sigma_\phi(x)}{1 + \nu^{-1}n}, \nu + n \right).$$

## 4.2 OBJECTIVE FUNCTION

Harnessing Eq. (5) and the defined prior and posterior distributions of the proposed C-$t^3$VAE, we derive in Appendix C the following class-wise objective

$$\mathcal{L}(\gamma, y) = \mathbb{E}_x \left[ \mathbb{E}_z \left[ (x - \mu_\theta(z))^\top \Sigma_x^{-1}(x - \mu_\theta(z)) \right] + (\mu_\phi(x) - \mu_y)^\top \Sigma_y^{-1}(\mu_\phi(x) - \mu_y) \right.$$

$$\left. + \frac{\nu \operatorname{Tr}\left( \Sigma_y^{-1} \Sigma_\phi(x) \right)}{\nu + n - 2} - \frac{\nu C_1}{C_2} |\Sigma_\phi(x)|^{-\frac{\gamma}{2(1+\gamma)}} \right],$$

with $C_1 = \left( C_{\nu+n,m}^\gamma \left( 1 + \frac{n}{\nu} \right)^{\frac{\gamma m}{2}} \frac{\nu+n+m-2}{\nu+n-2} \right)^{\frac{1}{1+\gamma}}$ and $C_2 = \left( \frac{C_{\nu,m+n}^\gamma}{|\Sigma_x|^{\frac{\gamma}{2}} |\Sigma_y|^{\frac{2\gamma+1}{2}}} \left( 1 + \frac{m+n}{\nu-2} \right)^{-\gamma} \right)^{\frac{1}{1+\gamma}}$.

By taking $\Sigma_x = \sigma^2 I$ and $\Sigma_y = I$, $\mathcal{L}(\gamma, y)$ objective function simplifies to :

$$\mathcal{L}(\gamma, y) = \mathbb{E}_x \left[ \frac{\mathbb{E}_z \left[ \|x - \mu_\theta(z)\|^2 \right]}{\sigma^2} + \|\mu_\phi(x) - \mu_y\|^2 + \frac{\nu \operatorname{Tr}\left( \Sigma_\phi(x) \right)}{\nu + n - 2} - \frac{\nu C_1}{C_2} |\Sigma_\phi(x)|^{-\frac{\gamma}{2(1+\gamma)}} \right]. \quad (10)$$

Therefore, we express the final loss function $\mathcal{L}(\gamma)$ over the whole dataset as : $\mathcal{L}(\gamma) = \sum_y \mathcal{L}(\gamma, y)$. As in Eq. (3), here too we consider all $p(y_i)$ to be equal and hence rule out their contribution to the loss function. This is done to avoid emphasizing the imbalance present in the data.

## 4.3 SAMPLING DISTRIBUTION

Similarly to the objective function of $t^3$VAE, $\mathcal{L}(\gamma, y)$ in Eq. (10) can be decomposed into a reconstruction term and regularization terms. To sample from the latent space of the C-$t^3$VAE, we focus on the regularization terms and define the following sampling distribution :

$$p_\nu^\star(z) = \sum_{y=1}^K \alpha_y \cdot t_m(\mu_y, \tau^2 I, \nu + n), \quad \forall y, \ \alpha_y = \frac{1}{K}. \quad (11)$$

The theoretical derivation of the variance $\tau^2$ leads to the form expressed in Eq. (9) (Derivation in Appendix D).

The mixture-based sampling distribution we define in Eq. (11) with equal $\alpha_y$ ensures a uniformly sampled synthetic data across all classes, regardless of their frequency in the original training data. As a result, C-$t^3$VAE equipped with this sampling distribution mitigates the common problem in generative models where head-class samples dominate due to their density in latent space. Furthermore, by modifying the mixture weights $\alpha_y$, one can prioritize specific classes. This makes our method flexible for targeted data augmentation or fairness-aware sampling strategies.

### 4.3.1 $\beta$-C-$t^3$VAE

As with $t^3$VAE, the class-wise objective defined in Eq. (10) can be split into a reconstruction and regularization terms. By preceding the regularization term with a $\beta$ scalar, we can define a $\beta$-C-$t^3$VAE model thereby improving the domain of applicability of the model.

Overall, C-$t^3$VAE provides a principled, flexible, and tractable framework for fair generative modeling, particularly under class-imbalanced conditions. In the following, we study the C-$t^3$VAE's performance across multiple datasets with varying imbalance degrees.

## 5 EXPERIMENTS

This section outlines the generative performance of the proposed C-$t^3$VAE model on labeled datasets compared to relevant VAE baselines[1]. We conduct experiments on three datasets notably SVHN-LT (Netzer et al., 2011), CIFAR100-LT (Krizhevsky, 2009; Cao et al., 2019) and CelebA (Liu et al., 2015) each chosen to highlight different challenges.

### 5.1 EVALUATION PROCEDURE

All models are evaluated using Fréchet Inception Distance (FID) (Heusel et al., 2017), computed against a *balanced* test set for each dataset. This setup measures how effectively a model overcomes training set imbalance by assessing its ability to generate high-quality samples across all classes. To evaluate robustness, we impose varying degrees of imbalance during training.

For SVHN-LT and CIFAR100-LT, we introduce class imbalance by applying an exponential decay to the number of samples per class after equalizing class sizes in the original train and test sets. The imbalance ratio $\rho$ defines the ratio between the most and least frequent classes, with class-wise sample counts $M_{y_i}$ given by: $M_{y_i} = M \cdot \rho^{-\frac{y_i-1}{K-1}}$, where $M$ is the original per-class sample count.

For CelebA, we compute FID per attribute, treating each attribute (eg. Mustache) and its negation (eg. no Mustache) as separate binary classes. The training set uses CelebA's inherent imbalance, while the test set is balanced by downsampling to the smaller class size. Multi-attribute generation is not considered in this work and is left for future exploration.

### 5.2 RESULTS

We present both quantitative and qualitative results of the C-$t^3$VAE model and the models it directly improves upon notably the VAE, C-VAE and $t^3$VAE models with their $\beta$ variants. This controlled comparison helps isolate the contributions of key design choices:

- **VAE :** ELBO trained standard Gaussian-based VAEs.

- **C-VAE :** VAE supplemented by conditional Gaussian priors to assess the class conditioning effect without changing the prior family.

- $t^3$**VAE :** Student's t-distribution latent prior and $\gamma$-power divergence objective, does not use class-conditional priors, nor allows for class conditional generation. Through a comparison with this model we assess the role of conditional modeling on Student's t-distribution priors.

---

[1]We restrict our comparisons to VAE-based generative models, as Latent Diffusion Models (Rombach et al., 2022), the state-of-the-art in image generation, depend on VAEs as a core component. Enhancing fairness in VAEs under class imbalance is therefore a necessary precursor to advancing more complex models.

In the following, we analyze the latent sampling standard deviation $\tau$ employed in $t^3$VAE and C-$t^3$VAE models, varying it to assess alignment between empirical and theoretical values. Then, we present FID comparisons across baselines with optimized hyper-parameters (The tuning of $\beta$, $\nu$ and $\tau$ hyper-parameters is reported in Appendix F), followed by per-class generative evaluation.

### 5.2.1 $\tau$ PARAMETER STUDY

Figure 1 illustrates the sampling standard deviation's $\tau$ impact on the FID metric. We observe that models based on the Student's t-distribution benefit from higher standard deviation on CIFAR100-LT compared to SVHN-LT. Specifically, C-$t^3$VAE outperforms the C-VAE for $\tau \in [0.25; 0.55]$ on CIFAR100-LT and $\tau \in [0.19; 0.28]$ on SVHN-LT and for all $\tau$ values on the CelebA dataset. Moreover, it surpasses the $t^3$VAE models' FID for all $\tau$ values and across all datasets.

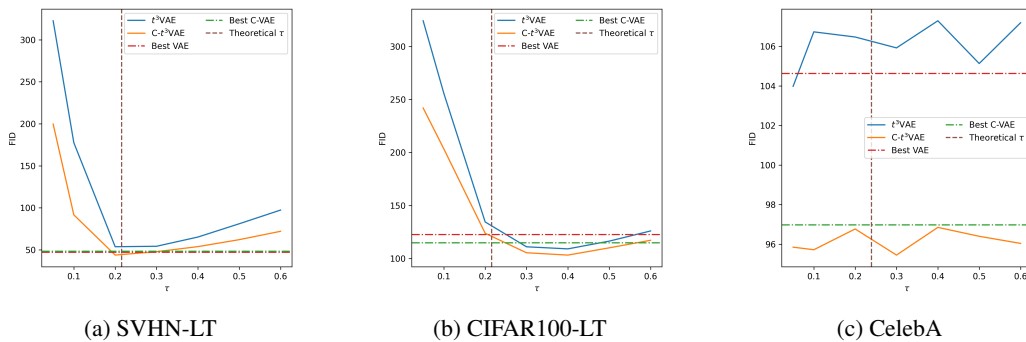

| (a) SVHN-LT | (b) CIFAR100-LT | (c) CelebA |

Figure 1: FID score as a function of $\tau$ for the $t^3$VAE and C-$t^3$VAE models. Results are for the imbalance ratio $\rho = 100$ for the SVHN-LT and CIFAR100-LT, and for the Mustache attribute ($\rho = 25$) in the case of the CelebA dataset. Other imbalance ratios' results paint a similar picture and are provided in Appendix F.3. The horizontal dashed lines is the FID value of the best performing VAE and C-VAE on each dataset and the vertical dashed line is the value of $\tau$ as derived in Eq. (9). We note that the used models in these figures have optimized $\beta$ and $\nu$ hyper-parameters.

For both Student's t-distribution based models, the optimal FID score for the SVHN-LT occurs near the theoretically derived $\tau$ value. However, for the more complex CIFAR100-LT dataset, the optimal $\tau$ is higher than the theoretical value $\tau = 0.4$. We hypothesize that the higher variance reflects the model's need during training to accommodate the greater complexity and entropy of the dataset, which makes it harder to align the learned posterior with the prior and to produce confident reconstructions. For the CelebA dataset, the standard deviation parameter $\tau$ has minimal to no impact on model performance likely due to the lower variability in the dataset's images as they all represent faces.

### 5.2.2 OPTIMIZED MODEL RESULTS DISCUSSION

After optimizing the hyperparameters of the various models tested in this work, we present their generation FID scores in Table 1. We provide results for $\beta$ models after optimization and non $\beta$ models ($\beta = 1$) to underscore the importance of this parameter on the resulting performance as it was not explored in the $t^3$VAE work (Kim et al., 2024).

From Table 1, $t^3$VAE consistently improves over the VAE model. This underlines the gen-

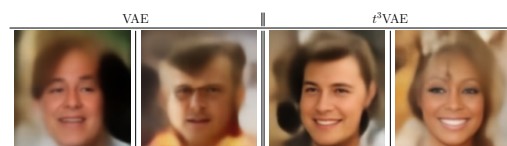

Figure 2: Sample synthetic images from the optimized VAE and $t^3$VAE models trained on the CelebA dataset. No class conditioning is possible for these models.

erative advantage of the Student's t-distribution prior over a Gaussian one, in addition to the reconstruction advantage highlighted by Kim et al. (2024). Moreover, optimizing the $\beta$ hyper-parameter improves $t^3$VAE's FID over the VAE on the CIFAR100-LT and CelebA, while remaining competitive on SVHN-LT. Also, qualitative results of Figure 2 show that the $t^3$VAE model compared to the VAE model produces significantly sharper synthetic images on the CelebA dataset.

Table 1: Generation FID results on the SVHN-LT, CIFAR100-LT and CelebA datasets. For the SVHN-LT and CIFAR100-LT datasets we use different imbalance ratios $\rho \in \{100, 50, 10, 1\}$. However, for the CelebA dataset we use the Mustache, Young, Male and Smiling attributes which have imbalance ratios of 25, 3.5, 1.4 and 1 respectively. The $\beta$ models undertook an optimization of the $\beta$ hyper-parameter while non-$\beta$ models have $\beta = 1$. All models have optimized $\nu$ and $\tau$ hyper-parameters. The attributes for the CelebA dataset column indicate which attribute is used to condition the conditional models and balance the test set.

| Models | SVHN-LT | | | | CIFAR100-LT | | | | CelebA | | | |
| | $\rho =100$ | 50 | 10 | 1 | $\rho =100$ | 50 | 10 | 1 | Mustache | Young | Male | Smiling |
|---|---|---|---|---|---|---|---|---|---|---|---|---|
| VAE | 93.89 | 91.91 | 91.66 | 92.16 | 163.66 | 162.91 | 165.47 | 166.46 | 110.58 | 92.01 | 110.58 | 82.05 |
| $\beta$-VAE | 47.11 | 49.81 | 45.70 | 43.48 | 122.62 | 123.07 | 123.72 | 124.43 | 104.63 | 92.87 | 87.96 | 83.15 |
| C-VAE | 74.75 | 70.40 | 72.30 | 74.16 | 157.90 | 163.67 | 162.09 | 163.24 | 96.98 | 89.17 | 86.17 | **78.35** |
| $\beta$-C-VAE | 48.39 | 46.39 | 43.97 | **43.87** | 114.88 | 118.89 | 114.89 | 118.21 | 98.35 | 85.53 | **79.76** | 78.46 |
| $t^3$VAE | 57.07 | 54.30 | 52.10 | 51.52 | 136.63 | 137.24 | 138.92 | 135.23 | 105.80 | 88.07 | 83.62 | 78.90 |
| $\beta$-$t^3$VAE | 51.62 | 49.55 | 48.93 | 45.37 | 109.11 | 107.93 | 108.97 | **111.00** | 105.86 | 88.21 | 83.83 | 78.89 |
| C-$t^3$VAE | 47.09 | 46.29 | 47.43 | 51.32 | 125.48 | 127.96 | 130.28 | 129.40 | 101.18 | 87.07 | 81.92 | 80.97 |
| $\beta$-C-$t^3$VAE | **44.02** | **42.60** | **42.01** | 44.49 | **103.25** | **102.99** | **105.92** | 112.37 | **95.82** | **82.61** | 81.65 | 80.08 |

For class-conditional models, $\beta$ optimized C-$t^3$VAE yields strong FID improvements across all imbalance settings, surpassing all baselines. As per Table 1, it achieves a gain of up to 4, 5 and 10 FID points over the $\beta$-$t^3$VAE model on the imbalanced settings of the SHVN-LT, CIFAR100-LT and CelebA, respectively. This underlines the importance of an equal per-class volume in an imbalanced dataset setting. Moreover, $\beta$-C-$t^3$VAE reduces the FID by up to 4 and 15 points over C-VAE on SVHN-LT and CIFAR100-LT respectively as per Table 1. For the CelebA dataset, Table 1 shows that $\beta$-C-$t^3$VAE achieves the best results on heavily imbalanced attributes like Mustache, demonstrating better generation for underrepresented groups. This gain in performance compared to the C-VAE goes back to the use of Student's t-distribution latent prior and its ability to better capture intra-class long-tail distributions. Additionally, qualitative samples of conditional optimized models (Figure 3) reveal sharper facial features in C-$t^3$VAE compared to the C-VAE. These results establish it as the most reliable model across imbalance ratios and resolutions.

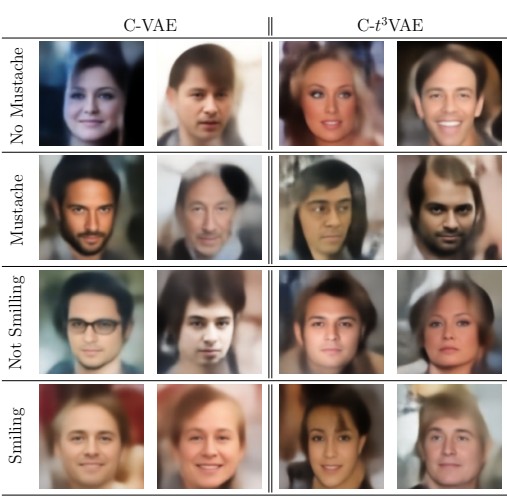

Figure 3: Sample synthetic images for the optimized C-VAE and C-$t^3$VAE trained on specific attributes of the CelebA dataset.

In summary, C-$t^3$VAE after optimization of all hyper-parameters notably $\beta$, $\nu$ and $\tau$ consistently achieves the lowest FID, validating the use of class-conditional Student's t-distribution priors and balanced latent sampling. Improvements on CIFAR100-LT are especially pronounced, highlighting the model's robustness to severe imbalance. However, for balanced settings, C-$t^3$VAE remains competitive.

### 5.2.3 PER-CLASS EVALUATION

In this section, we evaluate the conditional models on a per-class basis. However, since FID is biased on small datasets and offers limited insight as a single scalar metric, we rely on Precision, Recall, and F1 generative metrics[2] (Kynkäänniemi et al., 2019). Our results on CelebA are shown in Figure 4, with additional results for SVHN-LT and CIFAR100-LT included in Appendix G.

---

[2] Precision quantifies sample quality (sharpness), Recall reflects mode coverage and F1 is their harmornic mean. Together they allows us to gain deeper insight into the models' behavior.

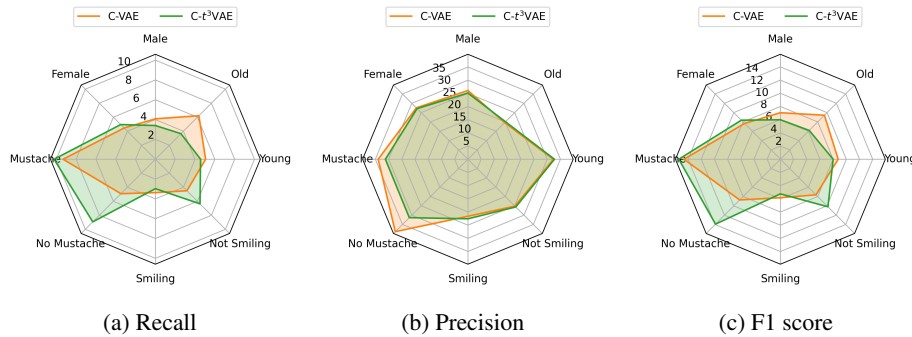

(a) Recall                (b) Precision                (c) F1 score

Figure 4: Per-class generative metrics on CelebA after optimization of all hyper-parameters notably $\beta$, $\nu$ and $\tau$. We note that the imbalance ratio of the Mustache, Young, Male and Smiling factors $\rho$ are 25, 3.5, 1.4 and 1 respectively.

On the CelebA dataset (Figure 4), C-$t^3$VAE improves Recall and F1 on the most imbalanced attribute (Mustache), but not on more balanced ones (Male, Smiling) which is an expected behavior. However, surprisingly, for the attribute Young (imbalance ratio 3.5), C-VAE performs slightly better this suggests that there is a regime where Gaussian priors may still suffice. To explore this, we vary the imbalance ratio on SVHN-LT from 100 to 1 and plot the results in Figure 5. This figure shows there is a threshold $\rho \approx 5$ before which C-VAE performs better; beyond that, C-$t^3$VAE has the advantage. Also, we notice that when $\rho \geq 5$, the more the imbalance ratio increases, the more FID gap becomes more significant and in favor of the C-$t^3$VAE.

For the SVHN-LT dataset (Figures in Appendix G), when comparing the C-$t^3$VAE to the C-VAE, the C-$t^3$VAE achieves significantly higher Recall and competitive Precision. This indicates a better coverage of data modes and maintaining images quality which yields higher F1 scores especially on tail classes. For the CIFAR100-LT dataset (Figures in Appendix G), C-VAE often achieves high Precision but zero Recall, indicating mode collapse. In contrast, the C-$t^3$VAE preserves Recall at the cost of slightly lower Precision, also resulting in improved F1 scores. Consequently, on the SVHN-LT CIFAR100-LT and CelebA datasets, C-$t^3$VAE sets itself as the most reliable method for fair, high quality image generation.

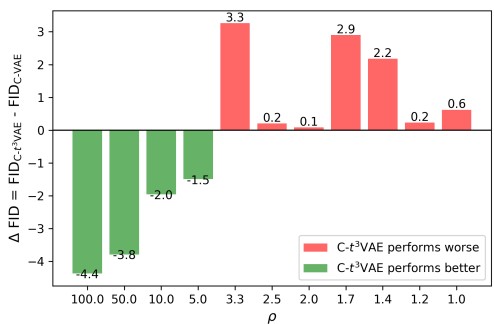

Figure 5: Fine-grained comparison of C-$t^3$VAE and C-VAE models under varying imbalance ratios on SVHN-LT.

## 6 CONCLUSION

We introduced C-$t^3$VAE, a class-conditional generative model that leverages Student's t-distributions in the latent space with a theoretically derived balanced sampling scheme. This approach improves fairness and sample quality under class imbalance. Extensive experiments on SVHN-LT, CIFAR100-LT, and CelebA demonstrate that, after the optimization of the $\beta$, $\nu$ and $\tau$ hyper-parameters, C-$t^3$VAE consistently outperforms the $t^3$VAE and conditional VAE baselines, achieving up to a 15-point FID improvement on highly imbalanced datasets. Per-class Precision, Recall, and F1 evaluations confirm improved mode coverage, particularly in tail classes. Moreover, we identify $\rho \geq 5$ as the regime where C-$t^3$VAE outperforms Gaussian-prior models, with the performance gap widening as $\rho$ increases. For $\rho < 5$, C-$t^3$VAE remains competitive with C-VAE.

Beyond generative performance, C-$t^3$VAE contributes to fairness-aware modeling by promoting balanced latent sampling and better mode coverage in underrepresented classes. Future work will explore extending C-$t^3$VAE to multi-label problems and integration with Latent Diffusion models.

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

# Appendix for
# "Conditional-$t^3$ VAE: Equitable Latent Space Allocation for Fair Generation"

## A $\gamma$-POWER DIVERGENCE CORRECTED DERIVATION

In this section, we present our correction of the previously derived form of the $\gamma$-power divergence in (Kim et al., 2024). We start from the integrals derived in (Kim et al., 2024) and keep a similar notation :

$$\int q(x)p(x)^\gamma dx = C^\gamma_{\nu,d}|\Sigma_1|^{-\frac{\gamma}{2}}\left(1 + \frac{1}{\nu-2}\operatorname{Tr}(\Sigma_1^{-1}\Sigma_0) + \frac{1}{\nu}(\mu_0-\mu_1)^\top\Sigma_1^{-1}(\mu_0-\mu_1)\right)$$

$$\int q(x)^{1+\gamma} dx = C^\gamma_{\nu,d}|\Sigma_0|^{-\frac{\gamma}{2}}\left(1 + \frac{d}{\nu-2}\right)$$

$$\int p(x)^{1+\gamma} dx = C^\gamma_{\nu,d}|\Sigma_1|^{-\frac{\gamma}{2}}\left(1 + \frac{d}{\nu-2}\right)$$

Combining these formulas gives a similar enthropy $\mathcal{H}_\gamma$ to the one derived in (Kim et al., 2024):

$$\mathcal{H}_\gamma(q) = -C^{\frac{\gamma}{1+\gamma}}_{\nu,d}|\Sigma_0|^{-\frac{\gamma}{2(1+\gamma)}}\left(1 + \frac{d}{\nu-2}\right)^{\frac{1}{1+\gamma}}.$$

However, the cross-entropy $\mathcal{C}_\gamma$ takes the following form :

$$\mathcal{C}_\gamma(q,p) = -C^{\frac{\gamma}{1+\gamma}}_{\nu,d}|\Sigma_0|^{\frac{\gamma^2}{2(1+\gamma)}}|\Sigma_1|^{-\frac{\gamma}{2}}\left(1 + \frac{1}{\nu-2}\operatorname{Tr}(\Sigma_1^{-1}\Sigma_0) + \frac{1}{\nu}(\mu_0-\mu_1)^\top\Sigma_1^{-1}(\mu_0-\mu_1)\right),$$

In red, we highlight the main difference to the formula derived in (Kim et al., 2024). $\mathcal{H}_\gamma(q)$ and $\mathcal{C}_\gamma(q,p)$ combine to give :

$$\mathcal{D}_\gamma(q\|p) = -\frac{C^{\frac{\gamma}{1+\gamma}}_{\nu,d}}{\gamma}\left(1 + \frac{d}{\nu-2}\right)^{-\frac{\gamma}{1+\gamma}}\left[-|\Sigma_0|^{-\frac{\gamma}{2(1+\gamma)}}\left(1 + \frac{d}{\nu-2}\right)\right.$$

$$\left. + |\Sigma_1|^{-\frac{\gamma}{2}}|\Sigma_0|^{\frac{\gamma^2}{2(1+\gamma)}}\left(1 + \frac{\operatorname{Tr}\left(\Sigma_1^{-1}\Sigma_0\right)}{\nu-2} + \frac{(\mu_0-\mu_1)^\top\Sigma_1^{-1}(\mu_0-\mu_1)}{\nu}\right)\right].$$

## B PRIORS DERIVATIONS

In this section, we present our derivations of the different prior distributions defining our proposed C-$t^3$-VAE model. Starting from the proposed joint distribution :

$$p_\theta(x,z|y) = \frac{C_{\nu,m+n}}{|\Sigma_x|^{\frac{1}{2}}|\Sigma_y|^{\frac{1}{2}}}\left[1 + \frac{(z-\mu_y)^\top\Sigma_y^{-1}(z-\mu_y)+(x-\mu_\theta(z))^\top\Sigma_x^{-1}(x-\mu_\theta(z))}{\nu}\right]^{-\frac{\nu+m+n}{2}}.$$

To calculate the prior distribution on the latent space we marginalize out $x$ as follows :

$$p(z|y) = \int p_\theta(x, z|y)dx$$

$$= \int C_{\nu,m+n}|\Sigma_x|^{-\frac{1}{2}}|\Sigma_y|^{-\frac{1}{2}}\left[1 + \frac{(z-\mu_y)^\top\Sigma_y^{-1}(z-\mu_y)}{\nu}\right.$$

$$\left. + \frac{(x-\mu_\theta(z))^\top\Sigma_x^{-1}(x-\mu_\theta(z))}{\nu}\right]^{-\frac{\nu+m+n}{2}}dx$$

$$= C_{\nu,m+n}|\Sigma_x|^{-\frac{1}{2}}|\Sigma_y|^{-\frac{1}{2}}\left[1 + \frac{1}{\nu}(z-\mu_y)^\top\Sigma_y^{-1}(z-\mu_y)\right]^{-\frac{\nu+m+n}{2}}$$

$$\times \int\left(1 + \frac{(1+\nu^{-1}m)(x-\mu_\theta(z))^\top\Sigma_x^{-1}(x-\mu_\theta(z))}{(1+\nu^{-1}(z-\mu_y)^\top\Sigma_y^{-1}(z-\mu_y))(\nu+m)}\right)^{-\frac{\nu+m+n}{2}}dx.$$

Given that :

$$\int C_{\nu+m,n}|\Sigma|^{-\frac{1}{2}}\left(1 + \frac{(x-\mu)^\top\Sigma^{-1}(x-\mu)}{\nu+m}\right)^{-\frac{\nu+m+n}{2}}dx = 1$$

$$\Rightarrow \int\left(1 + \frac{(x-\mu)^\top\Sigma^{-1}(x-\mu)}{\nu+m}\right)^{-\frac{\nu+m+n}{2}}dx = C_{\nu+m,n}^{-1}|\Sigma|^{\frac{1}{2}},$$

and when setting :

$$\Sigma^{-1} = \frac{(1+\nu^{-1}m)\Sigma_x^{-1}}{1+\nu^{-1}(z-\mu_y)^\top\Sigma_y^{-1}(z-\mu_y)},$$

We get :

$$\int\left(1 + \frac{(1+\nu^{-1}m)(x-\mu_\theta(z))^\top\Sigma_x^{-1}(x-\mu_\theta(z))}{(1+\nu^{-1}(z-\mu_y)^\top\Sigma_y^{-1}(z-\mu_y))(\nu+m)}\right)^{-\frac{\nu+m+n}{2}}dx$$

$$= C_{\nu+m,n}^{-1}\left|\left(\frac{(1+\nu^{-1}m)\Sigma_x^{-1}}{1+\nu^{-1}(z-\mu_y)^\top\Sigma_y^{-1}(z-\mu_y)}\right)^{-1}\right|^{\frac{1}{2}}$$

$$= C_{\nu+m,n}^{-1}\left|\frac{1+\nu^{-1}(z-\mu_y)^\top\Sigma_y^{-1}(z-\mu_y)}{(1+\nu^{-1}m)}\Sigma_x\right|^{\frac{1}{2}}$$

$$= C_{\nu+m,n}^{-1}\left(\frac{1+\nu^{-1}(z-\mu_y)^\top\Sigma_y^{-1}(z-\mu_y)}{1+\nu^{-1}m}\right)^{\frac{n}{2}}|\Sigma_x|^{\frac{1}{2}}.$$

Therefore, $p(z|y)$ simplifies to :

$$p(z|y) = C_{\nu,m+n}|\Sigma_x|^{-\frac{1}{2}}|\Sigma_y|^{-\frac{1}{2}}\left[1 + \frac{1}{\nu}(z-\mu_y)^\top\Sigma_y^{-1}(z-\mu_y)\right]^{-\frac{\nu+m+n}{2}}C_{\nu+m,n}^{-1}|\Sigma_x|^{\frac{1}{2}}$$

$$\times \left(\frac{1+\nu^{-1}(z-\mu_y)^\top\Sigma_y^{-1}(z-\mu_y)}{1+\nu^{-1}m}\right)^{\frac{n}{2}}$$

$$= C_{\nu,m+n}C_{\nu+m,n}^{-1}\left(1 + \frac{m}{\nu}\right)^{-\frac{n}{2}}|\Sigma_y|^{-\frac{1}{2}}\left[1 + \frac{1}{\nu}(z-\mu_y)^\top\Sigma_y^{-1}(z-\mu_y)\right]^{-\frac{\nu+m}{2}}$$

$$= C_{\nu,m}|\Sigma_y|^{-\frac{1}{2}}\left[1 + \frac{1}{\nu}(z-\mu_y)^\top\Sigma_y^{-1}(z-\mu_y)\right]^{-\frac{\nu+m}{2}}$$

$$= t_m(z|\mu_y, \Sigma_y, \nu).$$

Here and in the following, we use the fact

$$C_{\nu,m+n} = C_{\nu+m,n} C_{\nu,m} \left(1 + \frac{m}{\nu}\right)^{\frac{n}{2}}.$$

Besides, the prior distribution over the output of the decoder model $p(x|z,y)$ can be derived as follows :

$$p_\theta(x|z,y) = \frac{p_\theta(x,z|y)}{p(z|y)}$$

$$= \frac{C_{\nu,m+n}}{|\Sigma_x|^{\frac{1}{2}} |\Sigma_y|^{\frac{1}{2}}} \left[1 + \frac{(z-\mu_y)^\top \Sigma_y^{-1}(z-\mu_y) + (x-\mu_\theta(z))^\top \Sigma_x^{-1}(x-\mu_\theta(z))}{\nu}\right]^{-\frac{\nu+m+n}{2}}$$

$$\times C_{\nu,m}^{-1} |\Sigma_y|^{\frac{1}{2}} \left[1 + \frac{1}{\nu}(z-\mu_y)^\top \Sigma_y^{-1}(z-\mu_y)\right]^{\frac{\nu+m}{2}}$$

$$= C_{\nu+m,n} |\Sigma_x|^{-\frac{1}{2}} \left(1 + \frac{m}{\nu}\right)^{\frac{n}{2}} \left[1 + \frac{1}{\nu}(z-\mu_y)^\top \Sigma_y^{-1}(z-\mu_y)\right]^{-\frac{n}{2}}$$

$$\times \left(1 + \frac{(1+\nu^{-1}m)(x-\mu_\theta(z))^\top \Sigma_x^{-1}(x-\mu_\theta(z))}{\left(1 + \nu^{-1}(z-\mu_y)^\top \Sigma_y^{-1}(z-\mu_y)\right)(\nu+m)}\right)^{-\frac{\nu+m+n}{2}}$$

$$= t_n \left(x \middle| \mu_\theta(z), \frac{\left(1 + \nu^{-1}(z-\mu_y)^\top \Sigma_y^{-1}(z-\mu_y)\right)}{(1+\nu^{-1}m)} \Sigma_x, \nu+m\right).$$

## C  LOSS FUNCTION DERIVATION

In this section, we derive the loss function of C-$t^3$-VAE. We start by calculating the different double integrals $\iint p_\theta(x,z|y)^{1+\gamma} dx dz$, $\iint q_\phi(x,z|y) p_\theta(x,z|y)^\gamma dx dz$, and $\iint q_\phi(x,z|y)^{1+\gamma} dx dz$.

Firstly,

$$\iint p_\theta(x,z|y)^{1+\gamma} dx dz = \mathbb{E}_{z \sim p(z|y)} \mathbb{E}_{x \sim p_\theta(x|z,y)} \left[p_\theta(x,z|y)^\gamma\right]$$

$$= C_{\nu,m+n}^\gamma |\Sigma_x|^{-\frac{\gamma}{2}} |\Sigma_y|^{-\frac{\gamma}{2}} \mathbb{E}_z \mathbb{E}_x \left[1 + \frac{(z-\mu_y)^\top \Sigma_y^{-1}(z-\mu_y)}{\nu}\right.$$

$$\left. + \frac{(x-\mu_\theta(z))^\top \Sigma_x^{-1}(x-\mu_\theta(z))}{\nu}\right]$$

$$= C_{\nu,m+n}^\gamma |\Sigma_x|^{-\frac{\gamma}{2}} |\Sigma_y|^{-\frac{\gamma}{2}} \mathbb{E}_z \left[1 + \nu^{-1}(z-\mu_y)^\top \Sigma_y^{-1}(z-\mu_y)\right.$$

$$\left. + \nu^{-1} \mathbb{E}_x \left[\text{Tr}(\Sigma_x^{-1}(x-\mu_\theta(z))(x-\mu_\theta(z))^\top)\right]\right]$$

$$= C_{\nu,m+n}^\gamma |\Sigma_x|^{-\frac{\gamma}{2}} |\Sigma_y|^{-\frac{\gamma}{2}} \mathbb{E}_z \left[1 + \nu^{-1}(z-\mu_y)^\top \Sigma_y^{-1}(z-\mu_y)\right.$$

$$\left. + \nu^{-1} \text{Tr}\left(\Sigma_x^{-1} \Sigma_x \frac{\nu+m}{\nu+m-2} \frac{\left(1 + \nu^{-1}(z-\mu_y)^\top \Sigma_y^{-1}(z-\mu_y)\right)}{(1+\nu^{-1}m)}\right)\right]$$

Here, we use the following identities

$$(k-p)^\top H^{-1}(k-p) = \text{Tr}\left(H^{-1}(k-p)(k-p)^\top\right); \qquad \mathbb{E}[\text{Tr}(\cdot)] = \text{Tr}(\mathbb{E}[\cdot])$$

and the covariance of a multivariate Student's t-distribution $p \sim t(\mu; \Sigma; \nu)$ is $\frac{\nu}{\nu-2}\Sigma$. Consequently, and after a few simplifications we get

$$\iint p_\theta(x, z|y)^{1+\gamma} dxdz = C_{\nu,m+n}^\gamma |\Sigma_x|^{-\frac{\gamma}{2}} |\Sigma_y|^{-\frac{\gamma}{2}} \mathbb{E}_z \Bigg[ 1 + \nu^{-1}(z - \mu_y)^\top \Sigma_y^{-1}(z - \mu_y)$$

$$+ \frac{n}{\nu + m - 2} \left( 1 + \nu^{-1}(z - \mu_y)^\top \Sigma_y^{-1}(z - \mu_y) \right) \Bigg]$$

$$= C_{\nu,m+n}^\gamma |\Sigma_x|^{-\frac{\gamma}{2}} |\Sigma_y|^{-\frac{\gamma}{2}} \mathbb{E}_z \Bigg[ \left( 1 + \frac{n}{\nu + m - 2} \right)$$

$$\times \left( 1 + \nu^{-1}(z - \mu_y)^\top \Sigma_y^{-1}(z - \mu_y) \right) \Bigg]$$

$$= C_{\nu,m+n}^\gamma |\Sigma_x|^{-\frac{\gamma}{2}} |\Sigma_y|^{-\frac{\gamma}{2}} \left( 1 + \frac{n}{\nu + m - 2} \right)$$

$$\times \left( 1 + \nu^{-1}\mathbb{E}_z \left[ (z - \mu_y)^\top \Sigma_y^{-1}(z - \mu_y) \right] \right)$$

$$= C_{\nu,m+n}^\gamma |\Sigma_x|^{-\frac{\gamma}{2}} |\Sigma_y|^{-\frac{\gamma}{2}} \left( 1 + \frac{n}{\nu + m - 2} \right)$$

$$\times \left( 1 + \nu^{-1}\mathbb{E}_z \left[ \mathrm{Tr}(\Sigma_y^{-1}(z - \mu_y)(z - \mu_y)^\top) \right] \right)$$

$$= C_{\nu,m+n}^\gamma |\Sigma_x|^{-\frac{\gamma}{2}} |\Sigma_y|^{-\frac{\gamma}{2}} \left( 1 + \frac{n}{\nu + m - 2} \right) \left( 1 + \frac{m}{\nu - 2} \right)$$

$$= C_{\nu,m+n}^\gamma |\Sigma_x|^{-\frac{\gamma}{2}} |\Sigma_y|^{-\frac{\gamma}{2}} \left( 1 + \frac{m + n}{\nu - 2} \right).$$

Secondly,

$$\iint q_\phi(x, z|y) p_\theta(x, z|y)^\gamma dxdz = \mathbb{E}_{x \sim p_{data}} \mathbb{E}_{z \sim q(z|x)} \left[ p_\theta(x, z|y)^\gamma \right]$$

$$= C_{\nu,m+n}^\gamma |\Sigma_x|^{-\frac{\gamma}{2}} |\Sigma_y|^{-\frac{\gamma}{2}} \mathbb{E}_x \mathbb{E}_z \Bigg[ 1 + \frac{(z - \mu_y)^\top \Sigma_y^{-1}(z - \mu_y)}{\nu}$$

$$+ \frac{(x - \mu_\theta(z))^\top \Sigma_x^{-1}(x - \mu_\theta(z))}{\nu} \Bigg]$$

$$= C_{\nu,m+n}^\gamma |\Sigma_x|^{-\frac{\gamma}{2}} |\Sigma_y|^{-\frac{\gamma}{2}} \mathbb{E}_x \Bigg[ 1 + \frac{1}{\nu}\mathbb{E}_z \left[ (z - \mu_y)^\top \Sigma_y^{-1}(z - \mu_y) \right]$$

$$+ \frac{1}{\nu}\mathbb{E}_z \left[ (x - \mu_\theta(z))^\top \Sigma_x^{-1}(x - \mu_\theta(z)) \right] \Bigg].$$

Simplifying $\mathbb{E}_z \left[ (z - \mu_y)^\top \Sigma_y^{-1}(z - \mu_y) \right]$ :

$$\mathbb{E}_z \left[ (z - \mu_y)^\top \Sigma_y^{-1}(z - \mu_y) \right] = \mathbb{E}_z \left[ \mathrm{Tr} \left( \Sigma_y^{-1}(z - \mu_y)(z - \mu_y)^\top \right) \right]$$

$$= \mathbb{E}_z \left[ \mathrm{Tr} \left( \Sigma_y^{-1}(z - \mu(x) + \mu(x) - \mu_y)(z - \mu(x) + \mu(x) - \mu_y)^\top \right) \right]$$

$$= \mathbb{E}_z [\mathrm{Tr}(\Sigma_y^{-1}((z - \mu(x))(z - \mu(x))^\top + (z - \mu(x))(\mu(x) - \mu_y)^\top$$

$$+ (\mu(x) - \mu_y)(z - \mu(x))^\top + (\mu(x) - \mu_y)(\mu(x) - \mu_y)^\top))]$$

$$= \frac{\nu}{\nu + n - 2} \mathrm{Tr} \left( \Sigma_y^{-1}\Sigma_\phi(x) \right) + (\mu(x) - \mu_y)^\top \Sigma_y^{-1}(\mu(x) - \mu_y).$$

Then, $\iint q(x,z|y)p_\theta(x,z|y)^\gamma dxdz$ simplifies to :

$$\iint q_\phi(x,z|y)p_\theta(x,z|y)^\gamma dxdz = C_{\nu,m+n}^\gamma |\Sigma_x|^{-\frac{\gamma}{2}}|\Sigma_y|^{-\frac{\gamma}{2}}\mathbb{E}_x\left[1+\frac{1}{\nu}\mathbb{E}_z\left[(z-\mu_y)^\top\Sigma_y^{-1}(z-\mu_y)\right]\right.$$

$$\left.+\frac{1}{\nu}\mathbb{E}_z\left[(x-\mu_\theta(z))^\top\Sigma_x^{-1}(x-\mu_\theta(z))\right]\right]$$

$$\iint q_\phi(x,z|y)p_\theta(x,z|y)^\gamma dxdz = C_{\nu,m+n}^\gamma |\Sigma_x|^{-\frac{\gamma}{2}}|\Sigma_y|^{-\frac{\gamma}{2}}|\Sigma_y|^{-\frac{1}{2}}\mathbb{E}_x\left[1+\frac{1}{\nu}\frac{\nu\,\mathrm{Tr}\left(\Sigma_y^{-1}\Sigma_\phi(x)\right)}{\nu+n-2}\right.$$

$$+\frac{(\mu(x)-\mu_y)^\top\Sigma_y^{-1}(\mu(x)-\mu_y)}{\nu}$$

$$\left.+\frac{\mathbb{E}_z\left[(x-\mu_\theta(z))^\top\Sigma_x^{-1}(x-\mu_\theta(z))\right]}{\nu}\right].$$

Finally, the third term $\iint q_(x,z|y)^{1+\gamma}dxdz$ is

$$\iint q_\phi(x,z|y)^{1+\gamma}dxdz = C_{\nu+n,m}^\gamma\left(1+\frac{n}{\nu}\right)^{\frac{\gamma m}{2}}\left(1+\frac{m}{\nu+n-2}\right)\int|\Sigma_\phi(x)|^{-\frac{\gamma}{2}}p_{data}(x)^{1+\gamma}dx,$$

where this last double integral is equal to the one computed for the $t^3$-VAE.

Equipped with these formulas we can calculate the entropy $\mathcal{H}_\gamma$, cross-entropy $\mathcal{C}_\gamma$ and the $\gamma$-divergence $\mathcal{D}(q||p)$ of our model. Firstly,

$$\mathcal{H}_\gamma = -\left(\iint q(x,z)^{1+\gamma}dxdz\right)^{\frac{1}{1+\gamma}}$$

$$= -C_{\nu+n,m}^{\frac{\gamma}{1+\gamma}}\left(1+\frac{n}{\nu}\right)^{\frac{\gamma m}{2(1+\gamma)}}\left(1+\frac{m}{\nu+n-2}\right)^{\frac{1}{1+\gamma}}\left(\int|\Sigma_\phi(x)|^{-\frac{\gamma}{2}}p_{data}(x)^{1+\gamma}dx\right)^{\frac{1}{1+\gamma}},$$

Which is similar to the one calculated in the $t^3$VAE model.

Secondly,

$$\mathcal{C}_\gamma = -\left(\iint q(x,z|y)p_\theta(x,z|y)^\gamma dxdz\right)\left(\iint p_\theta(x,z|y)^{1+\gamma}\right)^{-\frac{\gamma}{1+\gamma}}$$

$$= -C_{\nu,m+n}^\gamma|\Sigma_x|^{-\frac{\gamma}{2}}|\Sigma_y|^{-\frac{\gamma}{2}-\frac{1}{2}}\mathbb{E}_x\left[1+\frac{1}{\nu}\frac{\nu\,\mathrm{Tr}\left(\Sigma_y^{-1}\Sigma_\phi(x)\right)}{\nu+n-2}+\frac{(\mu(x)-\mu_y)^\top\Sigma_y^{-1}(\mu(x)-\mu_y)}{\nu}\right.$$

$$\left.+\frac{1}{\nu}\mathbb{E}_z\left[(x-\mu_\theta(z))^\top\Sigma_x^{-1}(x-\mu_\theta(z))\right]\right]\left(C_{\nu,m+n}^\gamma|\Sigma_x|^{-\frac{\gamma}{2}}|\Sigma_y|^{-\frac{\gamma}{2}}\left(1+\frac{m+n}{\nu-2}\right)\right)^{-\frac{\gamma}{1+\gamma}}$$

$$= -\left(C_{\nu,m+n}^\gamma|\Sigma_x|^{\frac{-\gamma}{2}}|\Sigma_y|^{\frac{-\gamma}{2}}\right)^{\frac{1}{1+\gamma}}|\Sigma_y|^{-\frac{1}{2}}\left(1+\frac{m+n}{\nu-2}\right)^{\frac{-\gamma}{1+\gamma}}\mathbb{E}_x\left[1+\frac{1}{\nu}\left(\frac{\nu\,\mathrm{Tr}\left(\Sigma_y^{-1}\Sigma_\phi(x)\right)}{\nu+n-2}\right.\right.$$

$$\left.\left.+(\mu(x)-\mu_y)^\top\Sigma_y^{-1}(\mu(x)-\mu_y)+\mathbb{E}_z\left[(x-\mu_\theta(z))^\top\Sigma_x^{-1}(x-\mu_\theta(z))\right]\right)\right].$$

Hence, we can define our divergence as :

$$\mathcal{D}_\gamma(q\|p) = \frac{C_1}{\gamma}\left(\int |\Sigma_\phi(x)|^{-\frac{\gamma}{2}}p_{data}(x|y)^{1+\gamma}dx\right)^{\frac{1}{1+\gamma}} - \frac{C_2}{\gamma}\mathbb{E}_x\left[1 + \frac{1}{\nu}\left(\frac{\nu\,\mathrm{Tr}\left(\Sigma_y^{-1}\Sigma_\phi(x)\right)}{\nu+n-2}\right.\right.$$

$$\left.\left. + (\mu(x)-\mu_y)(\mu(x)-\mu_y)^\top + \mathbb{E}_z\left[(x-\mu_\theta(z))^\top\Sigma_x^{-1}(x-\mu_\theta(z))\right]\right)\right]$$

$$= \mathbb{E}_x\left[\frac{C_1}{\gamma}|\Sigma_\phi(x)|^{-\frac{\gamma}{2(1+\gamma)}} - \frac{C_2}{\gamma}\left(1 + \frac{1}{\nu}\left(\frac{\nu}{\nu+n-2}\,\mathrm{Tr}\left(\Sigma_y^{-1}\Sigma_\phi(x)\right)\right.\right.\right.$$

$$\left.\left.\left. + (\mu(x)-\mu_y)^\top\Sigma_y^{-1}(\mu(x)-\mu_y) + \mathbb{E}_z\left[(x-\mu_\theta(z))^\top\Sigma_x^{-1}(x-\mu_\theta(z))\right]\right)\right),\right]$$

with $C_1$ and $C_2$ being :

$$C_1 = C_{\nu+n,m}^{\frac{\gamma}{1+\gamma}}\left(1+\frac{n}{\nu}\right)^{\frac{\gamma m}{2(1+\gamma)}}\left(1+\frac{m}{\nu+n-2}\right)^{\frac{1}{1+\gamma}}$$

$$C_2 = \left(C_{\nu,m+n}^\gamma|\Sigma_x|^{-\frac{\gamma}{2}}|\Sigma_y|^{-\frac{2\gamma+1}{2}}\right)^{\frac{1}{1+\gamma}}\left(1+\frac{m+n}{\nu-2}\right)^{-\frac{\gamma}{1+\gamma}}.$$

On that account, the loss function for a class $y$ is :

$$\mathcal{L}(\gamma,y) = -\frac{\nu\gamma}{C_2}\cdot\mathcal{D}_\gamma(q\|p)$$

$$= \mathbb{E}_x\left[\mathbb{E}_z\left[(x-\mu_\theta(z))^\top\Sigma_x^{-1}(x-\mu_\theta(z))\right] + (\mu(x)-\mu_y)^\top\Sigma_y^{-1}(\mu(x)-\mu_y)\right.$$

$$\left. + \frac{\nu}{\nu+n-2}\,\mathrm{Tr}\left(\Sigma_y^{-1}\Sigma_\phi(x)\right) - \frac{\nu C_1}{C_2}|\Sigma_\phi(x)|^{-\frac{\gamma}{2(1+\gamma)}}\right],$$

and by taking $\Sigma_x = \sigma^2 I$ and $\Sigma_y = I$, we obtain :

$$\mathcal{L}_{(\gamma,y)} = \mathbb{E}_x\left[\frac{\mathbb{E}_z\left[||x-\mu_\theta(z)||^2\right]}{\sigma^2} + ||\mu(x)-\mu_y)||^2 + \frac{\nu\,\mathrm{Tr}\left(\Sigma_\phi(x)\right)}{\nu+n-2} - \frac{\nu C_1}{C_2}|\Sigma_\phi(x)|^{-\frac{\gamma}{2(1+\gamma)}}\right].$$

## D  SAMPLING DISTRIBUTION VARIANCE DERIVATION

In this section, we present the derivation of $\tau^2$ used in the sampling of $t^3$VAE and C-$t^3$VAE model. We present only the derivation for the C-$t^3$-VAE and it is identical to the one for the $t^3$-VAE since the former model is a generalization of the later.

First, we simplify the divergence $\mathcal{D}(q\|p^\star)$ :

$$\mathcal{D}(q\|p^\star) = -\frac{C_{\nu+n,m}^{\frac{\gamma}{1+\gamma}}}{\gamma}\left(1+\frac{m}{\nu+n-2}\right)^{-\frac{\gamma}{1+\gamma}}\left[-\left(1+\nu^{-1}n\right)^{\frac{\gamma m}{2(1+\gamma)}}|\Sigma_\phi(x)|^{-\frac{\gamma}{2(1+\gamma)}}\right.$$

$$\times\left(1+\frac{m}{\nu+n-2}\right) + |\tau^2 I|^{-\frac{\gamma}{2}}\left(1+\nu^{-1}n\right)^{-1}\Sigma_\phi(x)|^{\frac{\gamma^2}{2(1+\gamma)}}$$

$$\left.\times\left(1+\frac{\mathrm{Tr}\left(\tau^{-2}\left(1+\nu^{-1}n\right)^{-1}\Sigma_\phi(x)\right)}{\nu+n-2} + \frac{\tau^{-2}}{\nu+n}||\mu(x)-\mu_y||^2\right)\right]$$

Here, we use the fact that $|\alpha A|^\delta = \alpha^{\delta n} |A|^\delta$ where $n$ is the dimension of the square $A$ matrix. Also, we use $\text{Tr}(\alpha A) = \alpha \, \text{Tr}(A)$. After simplification and rearranging we get :

$$
\mathcal{D}(q\|p^\star) = -\frac{1}{\gamma} C_{\nu+n,m}^{\frac{\gamma}{1+\gamma}} \left(1 + \frac{m}{\nu+n-2}\right)^{-\frac{\gamma}{1+\gamma}} \left[ -\left(1+\nu^{-1}n\right)^{\frac{\gamma m}{2(1+\gamma)}} |\Sigma_\phi(x)|^{-\frac{\gamma}{2(1+\gamma)}} \right.
$$

$$
\times \left(1 + \frac{m}{\nu+n-2}\right) + \tau^{-\gamma m} \left(1+\nu^{-1}n\right)^{-\frac{\gamma^2 m}{2(1+\gamma)}} |\Sigma_\phi(x)|^{\frac{\gamma^2}{2(1+\gamma)}}
$$

$$
\left. \left(1 + \frac{\tau^{-2}\left(1+\nu^{-1}n\right)^{-1}}{\nu+n-2} \text{Tr}\left(\Sigma_\phi(x)\right) + \frac{\tau^{-2}}{\nu+n}\|\mu(x) - \mu_y\|^2\right)\right]
$$

$$
= -\frac{1}{\gamma} C_{\nu+n,m}^{\frac{\gamma}{1+\gamma}} \left(1 + \frac{m}{\nu+n-2}\right)^{-\frac{\gamma}{1+\gamma}} \left[ -\left(1+\nu^{-1}n\right)^{\frac{\gamma m}{2(1+\gamma)}} |\Sigma_\phi(x)|^{-\frac{\gamma}{2(1+\gamma)}} \right.
$$

$$
\times \left(1 + \frac{m}{\nu+n-2}\right) + \nu^{-1}\tau^{-2-\gamma m}\left(1+\nu^{-1}n\right)^{-\frac{\gamma^2 m}{2(1+\gamma)}-1} |\Sigma_\phi(x)|^{\frac{\gamma^2}{2(1+\gamma)}}
$$

$$
\left. \left(\kappa + \frac{\nu}{\nu+n-2} \text{Tr}\left(\Sigma_\phi(x)\right) + \|\mu(x) - \mu_y\|^2\right)\right]
$$

$$
= -\frac{1}{\gamma} C_{\nu+n,m}^{\frac{\gamma}{1+\gamma}} \left(1 + \frac{m}{\nu+n-2}\right)^{-\frac{\gamma}{1+\gamma}} \frac{\tau^{-2-\gamma m}}{\nu}\left(1+\nu^{-1}n\right)^{\frac{-\gamma^2 m}{2(1+\gamma)}-1} |\Sigma_\phi(x)|^{\frac{\gamma^2}{2(1+\gamma)}}
$$

$$
\times \left[ -\nu\tau^{2+\gamma m}\left(1+\nu^{-1}n\right)^{\frac{\gamma^2 m + \gamma m}{2(1+\gamma)}+1} |\Sigma_\phi(x)|^{\frac{-\gamma^2}{2(1+\gamma)}} |\Sigma_\phi(x)|^{\frac{-\gamma}{2(1+\gamma)}} \frac{\nu+n+m-2}{\nu+n-2} \right.
$$

$$
\left. + \kappa + \frac{\nu}{\nu+n-2} \text{Tr}\left(\Sigma_\phi(x)\right) + \|\mu(x) - \mu_y\|^2\right],
$$

with:

$$
\kappa = \nu\tau^2 \left(1+\nu^{-1}n\right).
$$

Then, we match the result to the loss function in Eq. (10) to get :

$$
\tau^{2+\gamma m}\left(1+\nu^{-1}n\right)^{\frac{\gamma m}{2}+1} |\Sigma_\phi(x)|^{-\frac{\gamma^2}{2(1+\gamma)}} \left(1 + \frac{m}{\nu+n-2}\right) = \frac{C_1}{C_2}.
$$

Moreover, we have :

$$
\frac{C_1}{C_2} = C_{\nu+n,m}^{\frac{\gamma}{1+\gamma}} \left(1 + \frac{n}{\nu}\right)^{\frac{\gamma m}{2(1+\gamma)}} \left(1 + \frac{m}{\nu+n-2}\right)^{\frac{1}{1+\gamma}} C_{\nu,m+n}^{-\frac{\gamma}{1+\gamma}} \sigma^{\frac{n\gamma}{1+\gamma}} \left(1 + \frac{m+n}{\nu-2}\right)^{\frac{\gamma}{1+\gamma}}
$$

$$
= \sigma^{\frac{n\gamma}{1+\gamma}} C_{\nu+n,m}^{\frac{\gamma}{1+\gamma}} C_{\nu,m+n}^{-\frac{\gamma}{1+\gamma}} \left(1 + \frac{n}{\nu}\right)^{\frac{\gamma m}{2(1+\gamma)}} \left(1 + \frac{m}{\nu+n-2}\right)^{\frac{1}{1+\gamma}} \left(1 + \frac{m+n}{\nu-2}\right)^{\frac{\gamma}{1+\gamma}}
$$

$$
= \sigma^{\frac{n\gamma}{1+\gamma}} C_{\nu,n}^{\frac{-\gamma}{1+\gamma}} \left(1 + \frac{m}{\nu+n-2}\right)^{\frac{1}{1+\gamma}} \left(1 + \frac{m+n}{\nu-2}\right)^{\frac{\gamma}{1+\gamma}}.
$$

Consequently we obtain :

$$
\tau^{2+\gamma m}\left(1+\nu^{-1}n\right)^{\frac{\gamma m}{2}+1} |\Sigma_\phi(x)|^{-\frac{\gamma^2}{2(1+\gamma)}} \left(1 + \frac{m}{\nu+n-2}\right)
$$

$$
= \sigma^{\frac{n\gamma}{1+\gamma}} C_{\nu,n}^{\frac{-\gamma}{1+\gamma}} \left(1 + \frac{m}{\nu+n-2}\right)^{\frac{1}{1+\gamma}} \left(1 + \frac{m+n}{\nu-2}\right)^{\frac{\gamma}{1+\gamma}}
$$

$$\tau^{2+\gamma m} = \frac{\sigma^{\frac{n\gamma}{1+\gamma}}}{C_{\nu,n}^{\frac{\gamma}{1+\gamma}}} \left(1+\nu^{-1}n\right)^{\frac{-\gamma m}{2}-1} |\Sigma_\phi(x)|^{\frac{\gamma^2}{2(1+\gamma)}} \left(1+\frac{m}{\nu+n-2}\right)^{-1} \left(1+\frac{m}{\nu+n-2}\right)^{\frac{1}{1+\gamma}}$$

$$\times \left(1+\frac{m+n}{\nu-2}\right)^{\frac{\gamma}{1+\gamma}}$$

$$= \frac{\sigma^{\frac{n\gamma}{1+\gamma}}}{C_{\nu,n}^{\frac{\gamma}{1+\gamma}}} \left(1+\nu^{-1}n\right)^{\frac{-\gamma m}{2}-1} |\Sigma_\phi(x)|^{\frac{\gamma^2}{2(1+\gamma)}} \left(1+\frac{m}{\nu+n-2}\right)^{-\frac{\gamma}{1+\gamma}} \left(1+\frac{m+n}{\nu-2}\right)^{\frac{\gamma}{1+\gamma}}$$

$$= \sigma^{\frac{n\gamma}{1+\gamma}} C_{\nu,n}^{\frac{-\gamma}{1+\gamma}} \left(1+\nu^{-1}n\right)^{-\frac{\gamma m}{2}-1} |\Sigma_\phi(x)|^{\frac{\gamma^2}{2(1+\gamma)}} \left(\frac{\nu+n-2}{\nu-2}\right)^{\frac{\gamma}{1+\gamma}}$$

$$= \left(1+\nu^{-1}n\right)^{-\frac{\gamma m}{2}-1} \left(\sigma^n C_{\nu,n}^{-1}|\Sigma_\phi(x)|^{\frac{\gamma}{2}} \frac{\nu+n-2}{\nu-2}\right)^{\frac{\gamma}{1+\gamma}}.$$

Hence, we get :

$$\tau^2 = \left(1+\nu^{-1}n\right)^{-\frac{\gamma m+2}{\gamma m+2}} \left(\sigma^n C_{\nu,n}^{-1}|\Sigma_\phi(x)|^{\frac{\gamma}{2}} \frac{\nu+n-2}{\nu-2}\right)^{\frac{2\gamma}{(1+\gamma)(2+\gamma m)}}$$

$$= \left(1+\nu^{-1}n\right)^{-1} \left(\sigma^n C_{\nu,n}^{-1}|\Sigma_\phi(x)|^{\frac{\gamma}{2}} \frac{\nu+n-2}{\nu-2}\right)^{\frac{2\gamma}{(1+\gamma)(2+\gamma m)}},$$

which is the form of $\tau^2$ we report in Eq. (9).

## E    EXPERIMENTAL SETUP DETAILS

### E.1    DATASETS

We conduct experiments on three datasets notably SVHN-LT (Netzer et al., 2011), CIFAR100-LT (Krizhevsky, 2009; Cao et al., 2019) and CelebA (Liu et al., 2015) each chosen to highlight different challenges related to generative modeling under class imbalance and varying visual complexity.

- **SVHN-LT :** This dataset is comprised of colored images of digits from 0 to 9 of size $32 \times 32 \times 3$. It serves as our controlled experimental setting. While simple enough for all models to converge, it is rich enough to reflect performance differences. However, as this dataset is naturally imbalanced, we balance the number of images across classes to have more control over the imposed imbalance ratio. In Table 2 we provide the number of images present in the dataset before balancing.

Table 2: The Number of images in the SVHN dataset for the train and test sets before balancing. The value in bold is the one used to balance the dataset.

| Class | 0 | 1 | 2 | 3 | 4 | 5 | 6 | 7 | 8 | 9 |
|---|---|---|---|---|---|---|---|---|---|---|
| Train set | 4948 | 13861 | 10585 | 8497 | 7458 | 6882 | 5727 | 5595 | 5045 | **4656** |
| Test set | 1744 | 5099 | 4149 | 2882 | 2523 | 2384 | 1977 | 2019 | 1660 | **1595** |

- **CIFAR100-LT :** This balanced dataset comprised of colored images of 100 classes of different natural objects. The images are of size $32 \times 32 \times 3$ presents high variability with 100 fine-grained classes. In the most extreme imposed imbalance setting, tail classes may contain as few as five examples. Though the images remain low-resolution, this setting stresses the models' robustness in the face of sparse data and various categories.
- **CelebA :** This dataset enables evaluation in real-world class-imbalance scenarios using attribute labels (e.g., Mustache, Young). It also introduces the complexity of higher resolution images ($178 \times 218 \times 3$), testing the scalability of our models. We select four binary attributes (Mustache, Young, Male, Smiling) for evaluation, with test sets balanced

by down-sampling the larger class. In Table 3, we provide the number of images for each chosen attribute in this dataset.

Table 3: The Number of images in the CelebA dataset for the train and test sets for the Mustache, Young, Male and Smiling attributes.

|  | Mustache | No Mustache | Young | Old | Male | Female | Smiling | Not Smiling |
|---|---|---|---|---|---|---|---|---|
| Train set | 6642 | 156128 | 126788 | 35982 | 68261 | 94509 | 78080 | 84690 |
| Test set | 722 | 19190 | 15114 | 4848 | 7715 | 12247 | 9987 | 9975 |

## E.2 DATA PREPROCESSING

- **SVHN-LT Dataset** For both training and testing, we crop each class to the minimum number of samples available across all classes. The only data augmentation applied is a random horizontal flip with 50% probability.
- **CIFAR100-LT Dataset** We use the dataset in its entirety without class filtering. As with SVHN-LT, we apply a random horizontal flip with 50% probability for data augmentation.
- **CelebA Dataset** We use the full dataset and apply a center crop of $160 \times 160$ to each image and then resize to $128 \times 128$. A random horizontal flip with 50% probability is also applied.

## E.3 MODEL ARCHITECTURE

Our encoder-decoder models follow a modular block design. Each encoder block consists of a convolutional layer, followed by 2D batch normalization and ReLU activation. Decoder blocks mirror this structure but replace convolutional layers with transposed convolutions.

- **SVHN-LT and CIFAR100-LT :** Encoders consist of four convolutional blocks with channels $\{64, 128, 256, 512\}$, followed by two linear layers for estimating mean and covariance. The decoder uses three transposed convolutional blocks with channel sizes $\{128, 64, 32\}$, ending with a three-channel convolution and Sigmoid activation.
- **CelebA :** The CelebA encoder includes six convolutional blocks with channels $\{64, 128, 256, 512, 512, 512\}$, ending with two linear layers. The decoder has six transposed convolutional layers with channels $\{512, 512, 256, 128, 64, 32\}$, followed by a final convolutional layer and Sigmoid activation.

## E.4 TRAINING DETAILS

All models are trained using the AdamW optimizer with a learning rate of $10^{-3}$ for 150 epochs. We use a batch size of 64 for SVHN-LT and CIFAR100-LT, and 128 for CelebA.

# F HYPERPARAMETER TUNING

We present the hyperparameter tuning process used across all evaluated models. We first optimize $\beta$, then $\nu$, and finally $\tau$, yielding the models' results reported in Table 1.

## F.1 $\beta$ OPTIMIZATION

We perform a hyperparameter study over $\beta$ for all tested models. Unless otherwise noted, we use the theoretically derived $\tau^2$ and set $\nu = 10$.

### F.1.1 ON THE SVHN-LT DATASET

As shown in Figure 6, the optimal $\beta$ values for Student's t-distribution based models lies in the range $\beta \in [0.4, 0.6]$ whereas it lies in the $\beta \in [0.05, 0.07]$ range for Gaussian-based models. This is because the regularization term in the $\gamma$-power divergence loss is ten times larger than the KL divergence. Figure 6 also shows that FID performance is highly sensitive to $\beta$ in the Gaussian setting, requiring careful tuning which is not the case for Student's t-distribution based models.

Finally, C-$t^3$VAE achieves the best FID surpassing the $t^3$VAE and the C-VAE for all imbalance settings.

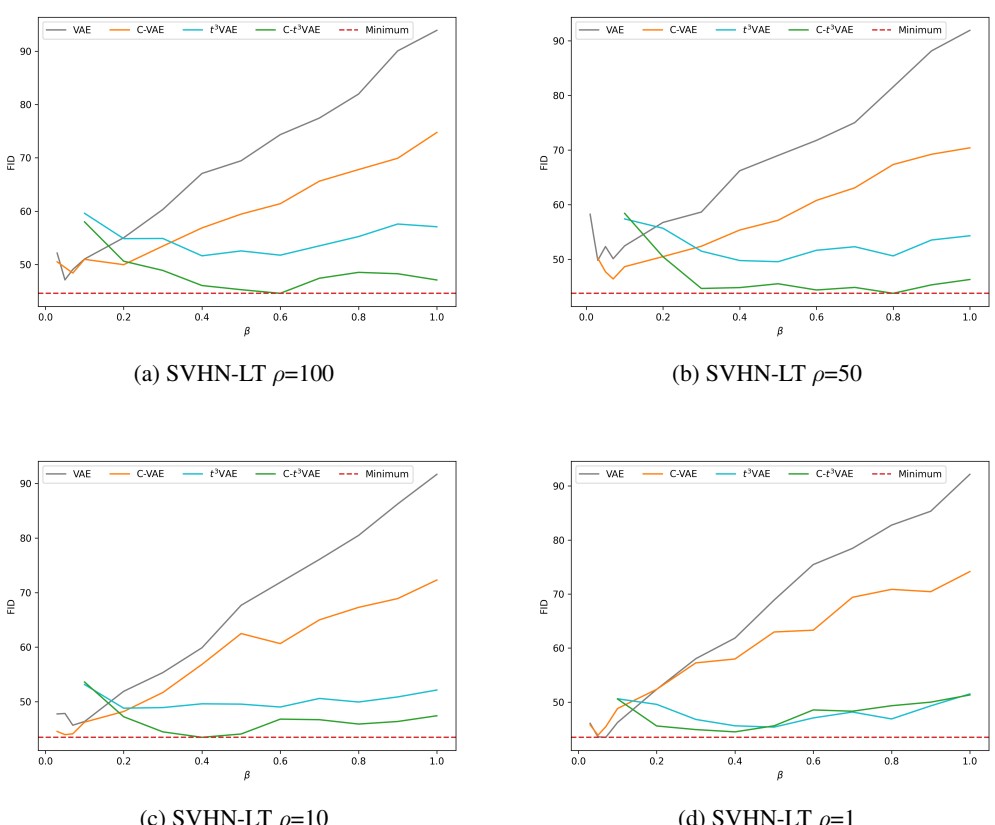

(a) SVHN-LT $\rho$=100

(b) SVHN-LT $\rho$=50

(c) SVHN-LT $\rho$=10

(d) SVHN-LT $\rho$=1

Figure 6: Variability of the FID as a function of the $\beta$ hyperparameter for the VAE, C-VAE, $t^3$VAE and C-$t^3$VAE on the SVHN-LT dataset.

### F.1.2 ON THE CIFAR100-LT DATASET

From Figure 7, we observe that Student's t-distribution based models obtain the best performance in terms of FID at $\beta = 0.2$ for CIFAR100-LT dataset. However, for the Gaussian-based models, the optimal value is much lower with $\beta \in [0.02, 0.05]$. The reason for this is that on this dataset too the KL regularization term is ten times smaller than the regularization terms present in the $\gamma$-power divergence loss. Additionally, we notice that C-VAE performs slightly better, likely due to the complexity of the dataset preventing full convergence to the imposed latent distribution. We further investigate this hypothesis in the $\tau$ analysis below.

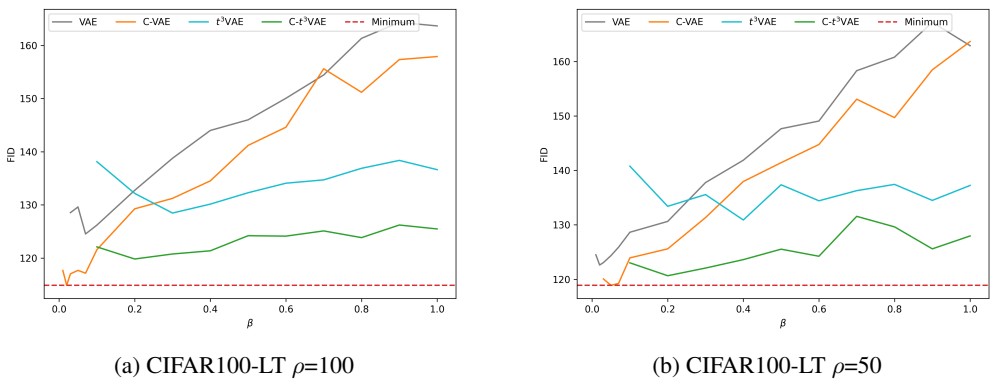

(a) CIFAR100-LT $\rho$=100

(b) CIFAR100-LT $\rho$=50

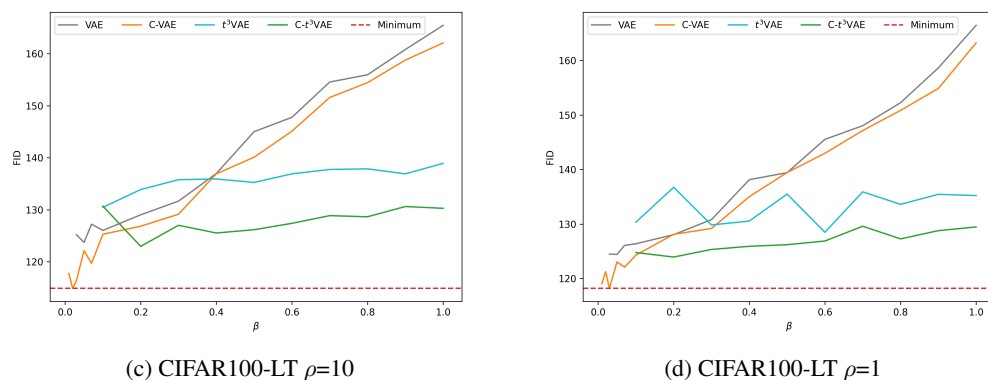

(c) CIFAR100-LT $\rho=10$        (d) CIFAR100-LT $\rho=1$

Figure 7: Variability of the FID as a function of the $\beta$ hyperparameter for the VAE, C-VAE, $t^3$VAE and C-$t^3$VAE on the CIFAR100-LT dataset.

### F.1.3 ON THE CELEBA DATASET

For the CelebA dataset, we focus only on $\beta$ optimization for Student's t-distribution based models. As for the Gaussian-based models, we set $\beta = 0.1$, because as shown in Table 1 the optimization of the $\beta$ hyperparameter has marginal impact on the FID for the CelebA dataset, which is not the case for the SVHN-LT and CIFAR100-LT dataset. Hence, we did not deem necessary to perform a hyper-parameter optimization over $\beta$ for the Gaussian based models. Nevertheless, from Figure 8 we see that $\beta$ has marginal effect on the Student's t-distribution based models too, likely due to the low intra-class variability.

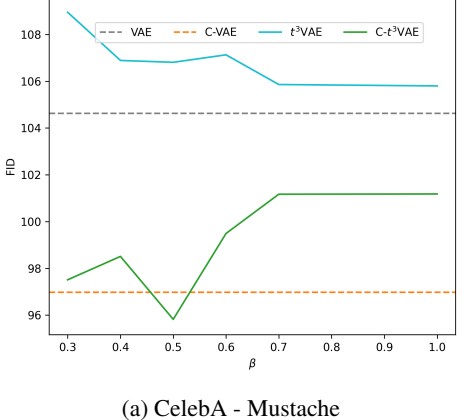 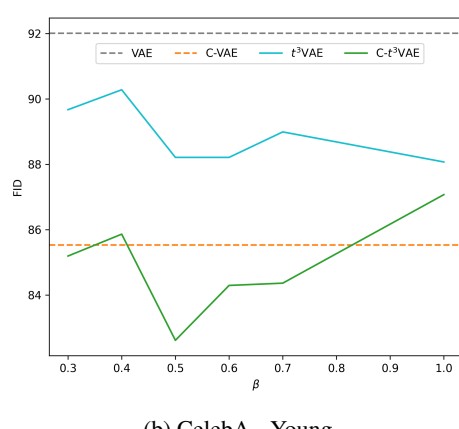

(a) CelebA - Mustache        (b) CelebA - Young

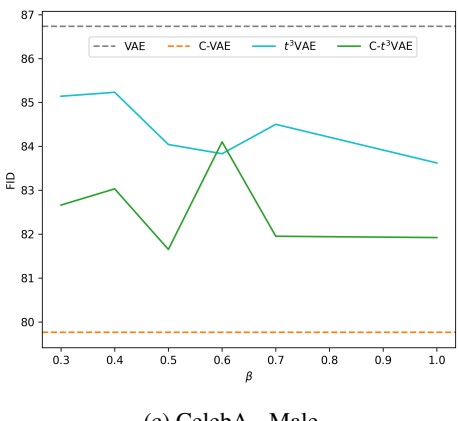 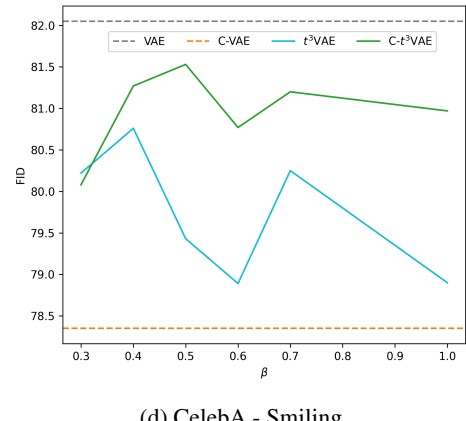

(c) CelebA - Male                    (d) CelebA - Smiling

Figure 8: Variability of the FID as a function of the $\beta$ hyperparameter for the $t^3$VAE and C-$t^3$VAE on the CelebA dataset. The horizontal lines for the VAE and C-VAE models are for the best performing model between $\beta = 0.1$ and $\beta = 1$.

## F.2  $\nu$ OPTIMIZATION

Table 4 shows results from tuning the degrees of freedom parameter $\nu$ in C-$t^3$VAE on both SVHN-LT and CIFAR100-LT across all imbalance ratios, using the optimal $\beta$ from the previous study. On average, $\nu = 10$ performs well, consistent with prior work (Kim et al., 2024). However, performance can be further improved by selecting $\nu$ within the range $[2.5, 20]$. Still, no major influence of hyper-parameter $\nu$ on the generative FID can be observed similarly to what was observed by Kim et al. (2024) for the reconstruction FID.

| $\nu$ | SVHN-LT | | | | CIFAR100-LT | | | |
|---|---|---|---|---|---|---|---|---|
| | 100 | 50 | 10 | 1 | 100 | 50 | 10 | 1 |
| 2.1 | 45.50 | 44.51 | 42.96 | 46.23 | 121.28 | 122.03 | 121.93 | 123.41 |
| 2.5 | 45.76 | 43.96 | 45.81 | 45.40 | 119.15 | 120.19 | **120.10** | 124.83 |
| 5 | 44.89 | **42.60** | 45.03 | 46.33 | 120.52 | 123.21 | 124.29 | 123.71 |
| 10 | 44.59 | 44.37 | 43.48 | **44.49** | **119.83** | 120.65 | 122.96 | **123.95** |
| 20 | **44.02** | 43.89 | **42.01** | 44.75 | 121.48 | **118.41** | 124.58 | 126.13 |
| 50 | 48.03 | 46.39 | 43.59 | 45.57 | **119.58** | 126.36 | 124.38 | 127.48 |
| 100 | 45.97 | 44.63 | 43.74 | 47.52 | 123.26 | 122.90 | 127.42 | 125.67 |

Table 4: Variability of the FID as a function of the standard deviation $\nu$ for the C-$t^3$VAE model on the SVHN-LT and CIFAR100-LT datasets.

## F.3  $\tau$ OPTIMIZATION

In this section, we evaluate the effect of the $\tau$ parameter on the SVHN-LT, CIFAR100-LT and CelebA datasets for all imbalance ratios while setting $\beta$ and $\nu$ to their previously optimized values. As shown in Figure 9, the optimal $\tau$ for SVHN-LT aligns closely with our theoretical prediction. In contrast, CIFAR100-LT consistently benefits from a larger $\tau = 0.4$, yielding improved FID across all imbalance settings and outperforming C-VAE. On CelebA, $\tau$ has minimal impact and the most likely value is $\tau \approx 0.3$. This value is between the value observed in for the SVHN-LT and CIFAR100-LT, as is mostly liked to the complexity and the entropy of the dataset where the more complex the data is the higher the value of $\tau$ is required.

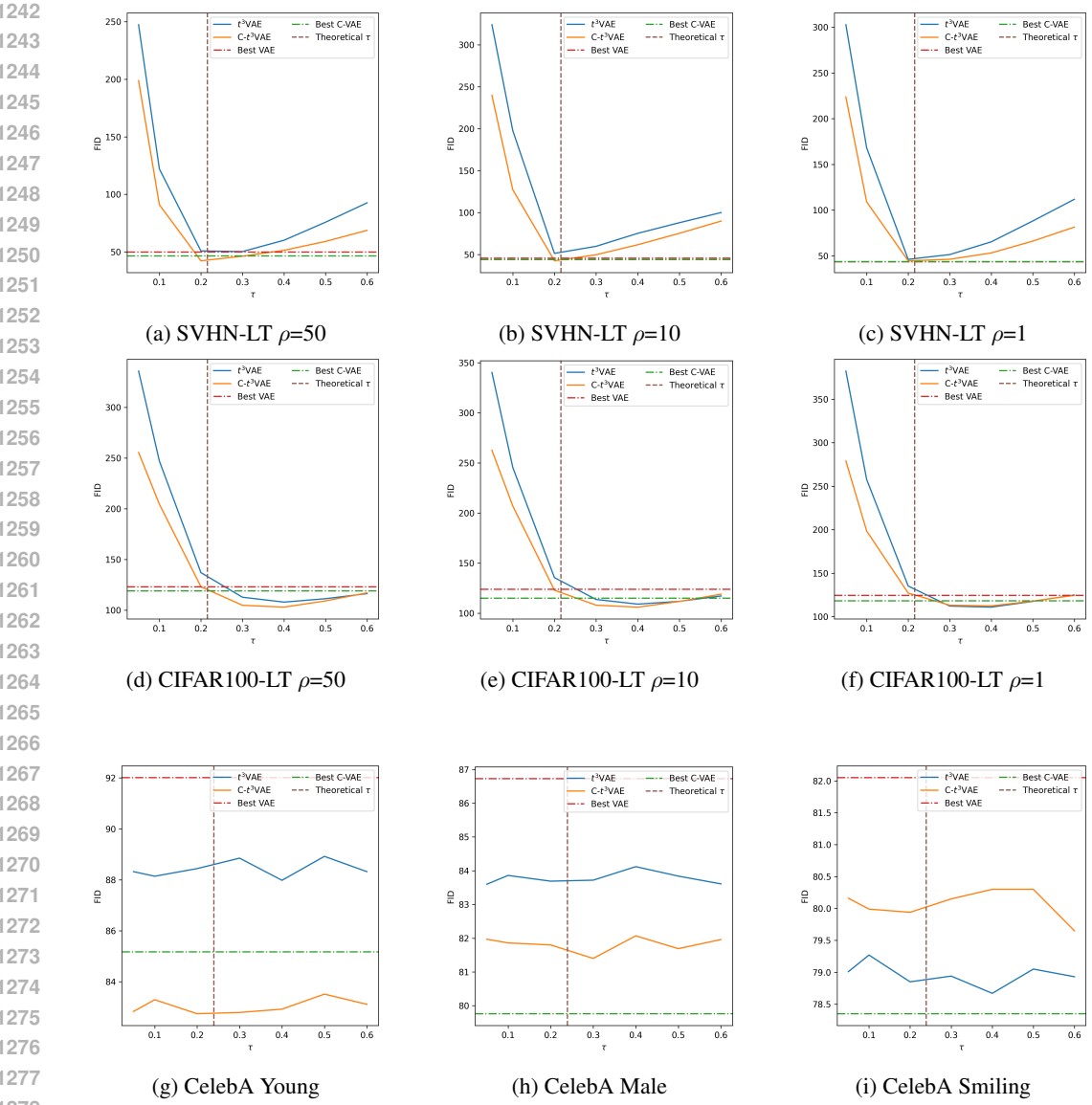

Figure 9: Variability of the FID as a function of the standard deviation $\tau^2$ for the $t^3$VAE and C-$t^3$VAE. In horizontal dashed lines is the FID value of the best performing VAE and C-VAE on each dataset. In vertical dashed lines is the theoretically identified value of $\tau$.

## G   PER-CLASS EVALUATION

In this section, we assess the conditional models' per-class Recall, Precision, and F1 metrics under all imbalance settings and for all tested datasets after optimization of all hyper-parameters.

From the following figures in Table 5 and 6, we see that the C-$t^3$VAE consistently improves Recall and mode coverage in highly imbalanced settings with $\rho = 100$ and $\rho = 50$. This comes at a minor Precision cost but results in significantly better F1 scores across most classes. However, on balanced or mildly imbalanced datasets, its performance remains competitive with Gaussian-based models. This observation is valid for both the SVHN-LT and CIFAR100-LT but is more pronounced on the later.

Table 5: Per-class generative metrics on SVHN-LT after optimization of $\beta$, $\nu$ and $\tau$ hyper-parameters.

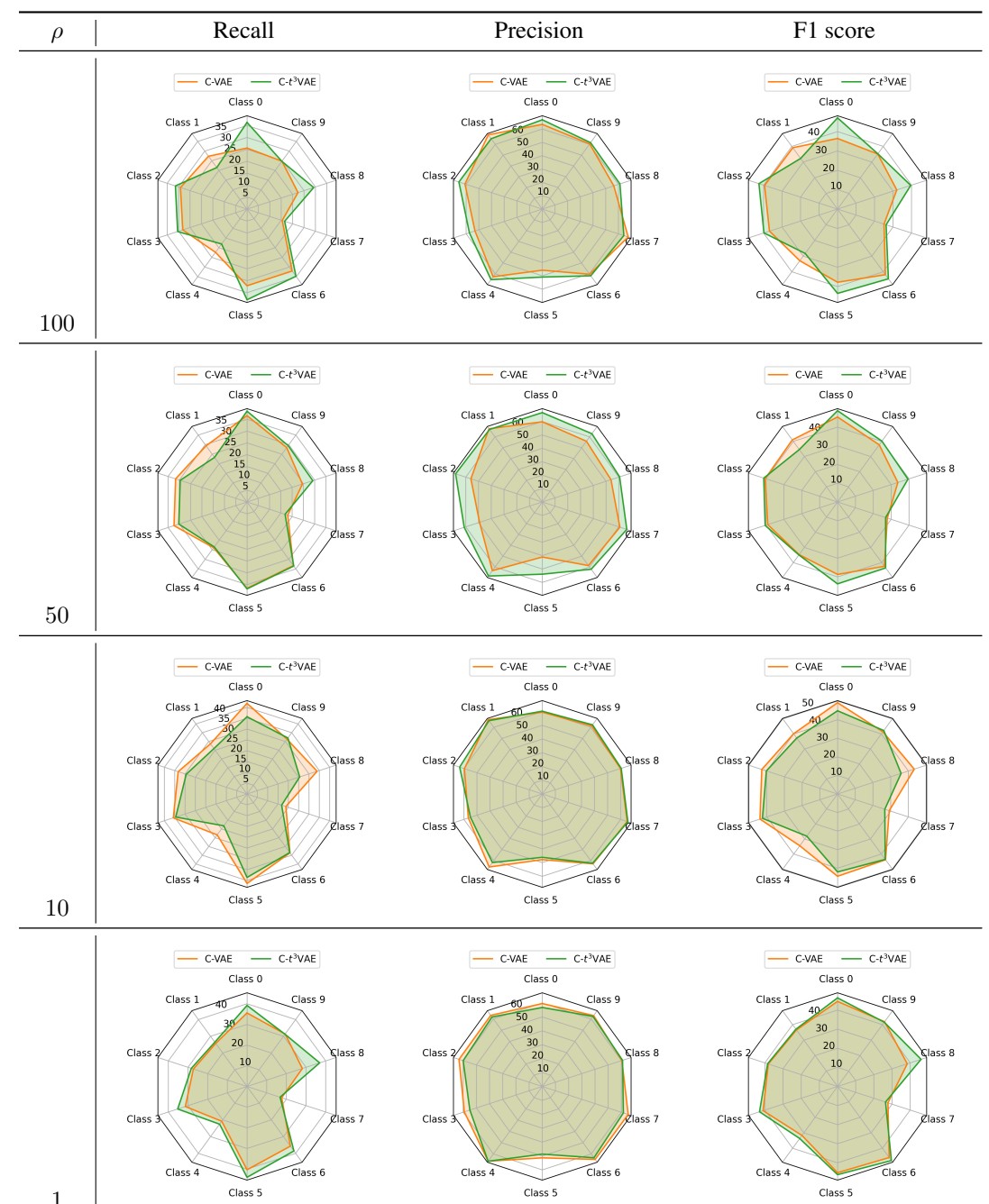

Table 6: Per-class generative metrics on CIFAR100-LT after optimization of $\beta$, $\nu$ and $\tau$ hyper-parameters, we focus on the top 5 head and tail classes.

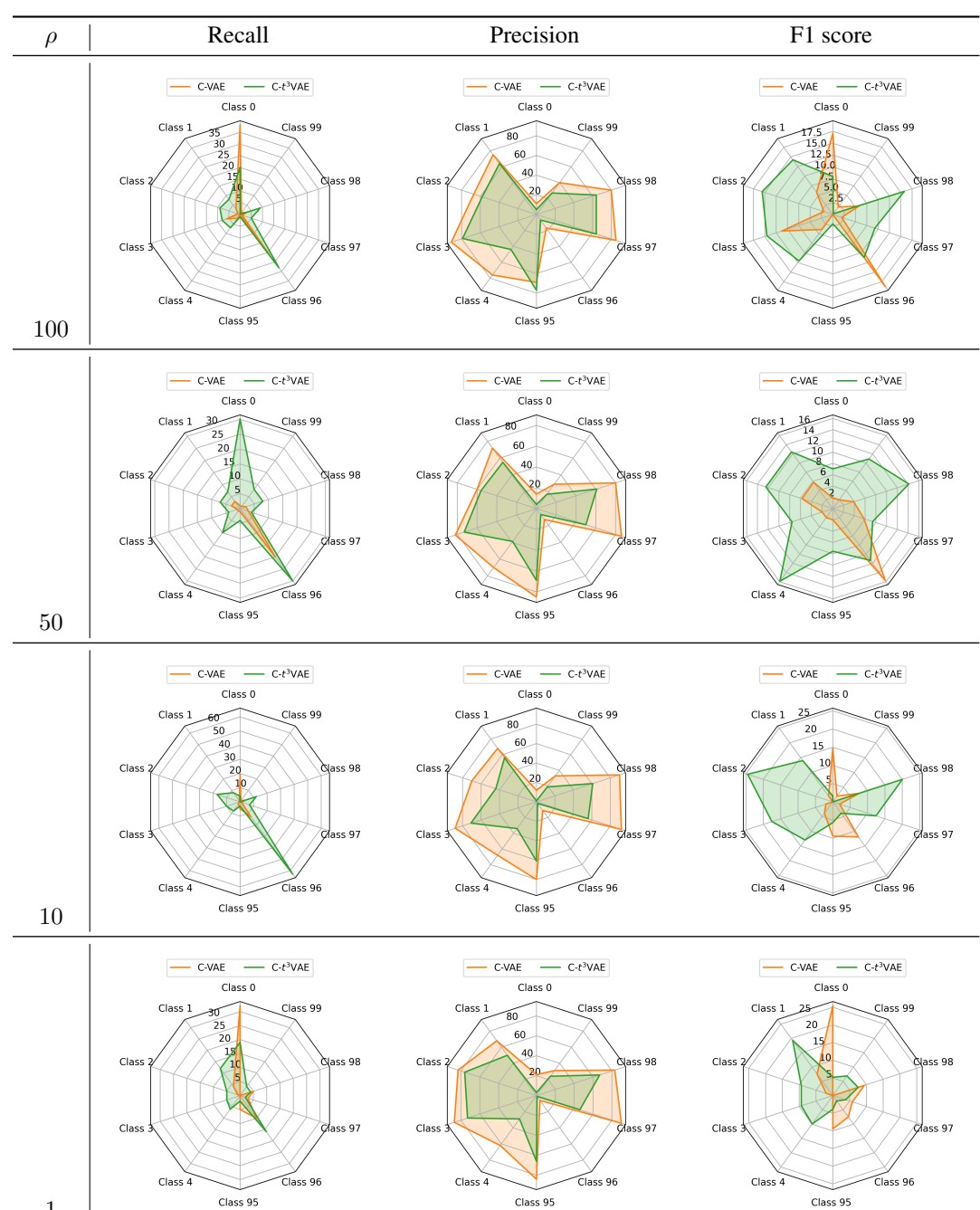

