# OpenReview forum: "Conditional-$t^3$VAE: Equitable Latent Space Allocation for Fair Generation"
_ICLR.cc/2026/Conference — Submitted to ICLR 2026_

### Official Review · Reviewer_V6dJ · 2025-10-21

**Soundness:** 3
**Presentation:** 3
**Contribution:** 3
**Rating:** 4
**Confidence:** 3

**Summary:**

The authors start by correctly claiming that t^{3} VAE is inadequate in addressing fair generation and equitable latent space allocation. The make the claim that a conditional VAE with per class heavy trailed Student's t-distribution priors can achieve this. They show promising results on the SVHN-LT, CIFAR100-LT, and CelebA datasets.

**Strengths:**

- The paper is well presented and is technically sound. The approach is simple and easy to understand.
- I like the intuition behind this, and in general class conditional makes sense for smaller scale trainings.
- The authors have correctly identified the core weakness behind the previous t^3 VAE for fair generation/ equitable latent space allocation.

**Weaknesses:**

- A weakness of the new optimization now compared to a simple t^{3} VAE is that class labels are required. With current day datasets in the scale of billions of samples, this is becoming more and more impractical.
- Furthermore, in the case of facial images, the set of attributes can explode very quickly. Race in itself is hard to constrain to a few classes. And each new subclass requires labelled examples.
- I would ideally like to see experimental results on larger scale higher quality datasets where both image quality and per class diversity can be ascertained. I don't think results on low quality datasets is sufficient especially when the evaluation metric is FID score.
- Further on using FID as a metric, how does the approach compare with t^{3} VAE on precision and recall as shown in Figure 4?
- Can you add results on a balanced dataset for reference?
- How does it perform on multi-attribute balancing?

**Questions:**

- How does this approach scale to billions of unlabelled examples? Would you first need a classifier to get the conditioning labels? Is there a way to make do without this?
- In line 306 the authors mention - "test set is balanced by downsampling to the smaller class size" what was the final sample size of this dataset? Is it sufficient to accurately measure FID score?

---

### Official Review · Reviewer_SdRH · 2025-10-29

**Soundness:** 2
**Presentation:** 3
**Contribution:** 1
**Rating:** 2
**Confidence:** 4

**Summary:**

This paper proposes Conditional $t^3$-VAE, an extension of the original $t^3$-VAE framework that introduces class conditioning to improve modeling performance on underrepresented classes. The authors reformulate the $t^3$-VAE’s $\gamma$-divergence objective under conditional settings and derive class-wise objectives and sampling rules accordingly. They demonstrate improved performance over VAE baselines on several class-imbalanced datasets.

**Strengths:**

- The paper focuses on an important problem of improving generative modeling under class imbalance.
- The writing is clear and easy to follow.

**Weaknesses:**

1. **Limited novelty.** The main contribution is the inclusion of conditioning within the existing $t^3$-VAE formulation for handling underrepresented classes, which is relatively straightforward. The mathematical derivations are largely mechanical extensions of those in prior works such as $t^3$-VAE and $\gamma$-VAE, with limited new theoretical development.

2. **Lack of meaningful baselines.** The experimental comparisons are restricted to VAE-type models, which are known to lag far behind modern diffusion-based generative models in both sample quality and diversity. Although the authors argue that VAEs remain useful for studying class imbalance, several diffusion-based methods have addressed the class imbalance issue (see [1] and [2]). To demonstrate the practical significance of this work, comparisons with diffusion-based class-imbalance approaches (like [1] and [2]) should be included.

3. **Evaluation metrics.** The paper mainly reports precision and recall, which are known to exhibit pathological behaviors. It is recommended to include improved metrics, such as density and coverage [3], to provide a more reliable evaluation of generative performance.

----
**References**

[1] Class-Balancing Diffusion Models, CVPR 2023

[2] Heavy-Tailed Diffusion Models, ICLR 2025

[3] Reliable Fidelity and Diversity Metrics for Generative Models, ICML 2020

**Questions:**

See weaknesses.

---

### Official Review · Reviewer_JGTj · 2025-10-30

**Soundness:** 3
**Presentation:** 4
**Contribution:** 2
**Rating:** 4
**Confidence:** 3

**Summary:**

The authors introduces conditional $t^3$VAE, which enables conditional generation of data given a class label $y$ by leveraging the $t^3$VAE model, a VAE model suitable for data with heavy-tail distribution. While the original $t^3$VAE applies multivariate Student's t-distributions for the joint prior of data $x$ and latent variable $z$, the authors introduce the joint prior distribution $p_\theta(x,z|y)$ with learnable mean vector $\mu_y$ for each class $y$, which is the Student's t-distribution. The sampling distribution for $p(z)$ is defined as a mixture of t-distributions with different means for each class, where equal weights are given to overcome the imbalance of the class. The performance of the $t^3$VAE is presented with SVHN-LT, CIFAR100-LT, and CelebA datasets. All datasets were trained with a class information where imbalance on the class size was present.

**Strengths:**

- C-$t^3$VAE enables conditional generation by incorporating the class variable $y$ into the existing $t^3$VAE. The authors successfully demonstrates its trainability, and the generated data samples conditioned on each class $y$ possess the characteristic features expected of their respective classes. Also, the proposed method demonstrated the best performance in scenarios where the training set exhibits extreme class imbalance, compared with other types of VAE.

- For the $\tau^2$ parameter used in the prior distribution of $z$, the authors provide improved value of $\tau^2$ consistent with the established theory.

**Weaknesses:**

- The results on per-class F1 scores presented in the Appendix are hard to interpret. For SVHN-LT and CIFAR100-LT, there are few cases where the C-VAE exhibits better metrics for minority classes, which contradictory with the paper's claim that C-$t^3$VAE exhibits better performance on the classes with imbalance.

- While the foundational motivation of $t^3$VAE is to explore the utility of heavy-tailed distribution within VAE framework, the experiment section of the paper is predominantly focused on comparing generation quality metrics (FID and the generative F1 score). It would be better if an additional quantitative or qualitative analysis about comparing the encoded latent representations to the conditional prior distribution on $z$ (e.g., using an MMD test or visualization of the latent space) is presented for enhancing the proposed model's theoretical significance.

**Questions:**

1. The covariance for the prior distribution for $z$ were all fixed to $\tau^2 I$ for every $y$. Would it be possible to also substitute the covariance into a term with a learnable parameter for each class, such as $\sigma_y^2 \tau^2 I$?

1. How will the metrics change if the class distribution of the test set does not matches with the prior distribution on the class $p(y)$? You can check this immediately by using all the test data from CelebA and not downsampling it to make it balanced.

1. For all the models compared, how does the test reconstruction loss differ?

---

### Official Review · Reviewer_ttdG · 2025-11-04

**Soundness:** 2
**Presentation:** 3
**Contribution:** 2
**Rating:** 4
**Confidence:** 4

**Summary:**

This paper considers an t^3VAE-based generative model for class-imbalanced distributions. The t^3VAE is a non-ELBO-based variational autoencoder model that employs a multivariate version of Student's t distribution as prior, encoder, and decoder to robustly model the heavy-tailed distributions. However, the t prior does not allocate latent space equally for each class. To address this problem, this paper proposes to use a t prior separately for each class. Each class possesses the loss function of the t^3VAE, which is summed over all classes with equal weights to yield the final loss function. The resulting model is called the C-t^3VAE. For generation, a t mixture with equal class-conditional probabilities is used. Empirical evidences show that the proposed C-t^3VAE enjoys improved FID and per-class evaluation metrics including the recall and F1 score.

**Strengths:**

The methodology is easy to understand. The emphasis in downstream tasks is on the generation performance, which is not extensively treated in the original t^3VAE. The improvement in generative performance in impressive. The reported phase transition phenomena from the conventional VAE to t^3VAE as a function of class imbalance ratio is also interesting.

**Weaknesses:**

1. Although the method is easy to follow, its approach largely follows that of the conditional VAE and therefore a straightforward extension of the t^3VAE. Also, unlike what the title suggests, virtually no treatment on fairness task is given in the paper.

2. In generation, the mixture weights $\alpha_y$ in $p_{\nu}^{*}(z)$ are fixed at $1/K$. Although it is noted that class prioritization can be achieved by adjusting $\alpha_k$, no experiment is conducted in this regard.

3. In the CelebA experiment, only binary conditioning is applied and multi-attribute generation is not considered.

4. The "correction" of the variance parameter $\tau^2$ in (9) for the alternative prior in t^3VAE is, unfortunately, incorrect. This stems from that the derivation of the cross entropy in Appendix A computes it by
$$
-\frac{\int q(x)p(x)^{\gamma}dx}{\left(\int q(x)^{1+\gamma}dx\right)^{\frac{\gamma}{1+\gamma}}}
$$
instead of
$$
-\frac{\int q(x)p(x)^{\gamma}dx}{\left(\int p(x)^{1+\gamma}dx\right)^{\frac{\gamma}{1+\gamma}}}
.
$$
See the definition of the gamma-power divergence in Kim et al. (2024, Eq. 8). The above mistake is likely from the typo in Proposition 3 of Kim et al. (2024): the integral $\int q(x)^{1+\gamma}dx$ in the third-to-last line of the proof should be $\int p(x)^{1+\gamma}dx$. Nevertheless, it is the correct $\tau^2$ ("approximation" in (9)) that is used in the computation, so the effect of the incorrectness in contained.

**Questions:**

1. Conditional VAE can be understood as maximizing conditional likelihood instead of the full likelihood. t^3VAE minimizes the gamma-power divergence between two statistical manifolds. How can the conditional t^3VAE be interpreted in terms of divergence minimization? If all the classes are equally likely, then minimizing the *single* gamma-power divergence between the two statistical manifolds defined by $p_{data}(x)q_{\phi}(z|x)$ and $\sum_{y} p(x, z | y) p(y)$ result in the objective function (10)?

2. I understand that all CelebA experiments are conducted with binary coding for each attribute, without considering multi-attribute conditioning. In that case, to reproduce the results in Table 1, were four separate experiments required? Or is $y$ treated as a 4-dimensional one-hot vector (i.e., considering 16 classes)? If the former is true, how about considering the latter approach? If the latter is true, how about comparing the results for multi-attribute generation?

---

### Meta-Review · Area_Chair_KzfJ · 2025-12-17

**Summary:**

While the reviewers agree the paper introduces a t^3VAE-based generative model for class-imbalanced distributions and is easy to follow, there are also significant concerns on the novelty, technical details, experiments, analysis, and related work. All the reviewer scores are on the negative side, and the authors did not post any rebuttals, which means that they agree with the reviews. This is a clear signal for rejecting the paper this time, and we hope that the comments can help further improve the paper.

**Reviewer Concerns:**

The authors did not respond to the reviews.

**Reviewer Scores:**

The reviewers would not have changed their scores as there was no discussion.

---

### Decision · Program_Chairs · 2026-01-26

Reject